# XDO: A Double Oracle Algorithm for Extensive-Form Games

**Stephen McAleer**
Department of Computer Science
University of California, Irvine
`smcaleer@uci.edu`

**John Lanier**
Department of Computer Science
University of California, Irvine
`jblanier@uci.edu`

**Kevin A. Wang**
Department of Computer Science
University of California, Irvine
`kevinwang@kevinwang.us`

**Pierre Baldi**
Department of Computer Science
University of California, Irvine
`pfbaldi@ics.uci.edu`

**Roy Fox**
Department of Computer Science
University of California, Irvine
`royf@uci.edu`

## Abstract

Policy Space Response Oracles (PSRO) is a reinforcement learning (RL) algorithm for two-player zero-sum games that has been empirically shown to find approximate Nash equilibria in large games. Although PSRO is guaranteed to converge to an approximate Nash equilibrium and can handle continuous actions, it may take an exponential number of iterations as the number of information states (infostates) grows. We propose Extensive-Form Double Oracle (XDO), an extensive-form double oracle algorithm for two-player zero-sum games that is guaranteed to converge to an approximate Nash equilibrium *linearly* in the number of infostates. Unlike PSRO, which mixes best responses at the root of the game, XDO mixes best responses at every infostate. We also introduce Neural XDO (NXDO), where the best response is learned through deep RL. In tabular experiments on Leduc poker, we find that XDO achieves an approximate Nash equilibrium in a number of iterations an order of magnitude smaller than PSRO. Experiments on a modified Leduc poker game and Oshi-Zumo show that tabular XDO achieves a lower exploitability than CFR with the same amount of computation. We also find that NXDO outperforms PSRO and NFSP on a sequential multidimensional continuous-action game. NXDO is the first deep RL method that can find an approximate Nash equilibrium in high-dimensional continuous-action sequential games. Experiment code is available at `https://github.com/indylab/nxdo`.

## 1 Introduction

Policy Space Response Oracles (PSRO) (Lanctot et al., 2017) is a reinforcement learning (RL) method for finding approximate Nash equilibria (NE) in large two-player zero-sum games. Methods based on PSRO have recently achieved state-of-the-art performance on large imperfect-information two-player zero-sum games such as Starcraft (Vinyals et al., 2019) and Stratego (McAleer et al., 2020). One major benifit of PSRO versus other deep RL methods for two-player zero-sum games is that it is naturally compatible with games that have continuous actions. The only other deep RL method

35th Conference on Neural Information Processing Systems (NeurIPS 2021).

compatible with continuous actions, self play, is not guaranteed to converge to a Nash equilibrium even in small games like Rock Paper Scissors. Despite the empirical success of PSRO, in the worst case, PSRO may need to expand all pure strategies in the normal form of the game, which grows exponentially in the number of information states (infostates). The reason for this is that PSRO is based on the Double Oracle algorithm for normal-form games (McMahan et al., 2003), and a mixture of normal-form pure strategies is an inefficient representation of extensive-form policies.

In this work, we propose a new double oracle algorithm, Extensive-Form Double Oracle (XDO), that is designed for extensive-form (sequential) games. Like PSRO, XDO keeps a population of pure strategies. At every iteration, XDO creates a restricted game by only considering actions that are chosen by at least one strategy in the population. This restricted game is then approximately solved via an extensive-form game solver, such as Counterfactual Regret Minimization (CFR) (Zinkevich et al., 2008) or Fictitious Play (FP) (Brown, 1951), to find a meta-NE, which is extended to the full game by taking arbitrary actions at infostates not encountered in the restricted game. Next, a best response (BR) to the restricted game meta-NE is computed and added to the population. XDO can be viewed as a version of PSRO where the restricted game allows mixing population strategies not only at the root of the game, but at every infostate.

XDO is guaranteed to converge to an approximate NE in a number of iterations that is linear in the number of infostates, while PSRO may require a number of iterations exponential in the number of infostates. Furthermore, on a worst-case family of games for the lower bound on the number of PSRO iterations, we show that XDO converges in a number of iterations that does not grow with the number of infostates, and grows only linearly with the number of actions at each infostate.

We also introduce a neural version of XDO, called Neural XDO (NXDO). NXDO can be used in games that are large enough to benefit from the generalization over infostates induced by neural-network strategies. NXDO learns approximate BRs through any deep reinforcement learning algorithm. The restricted game consists of meta-actions, each selecting a population policy to play the next action. This restricted game is then solved through any neural extensive-form game solver, such as NFSP (Heinrich & Silver, 2016) or Deep CFR (Brown et al., 2019). In our experiments, we use PPO (Schulman et al., 2017) or DDQN (Van Hasselt et al., 2016) for the approximate BR and NFSP as the restricted game solver. Although convergence guarantees may not apply in such cases, like PSRO, NXDO is compatible with continuous action spaces.

In games with a large number of actions, NXDO and PSRO effectively prune the game tree and outperform methods such as Deep CFR and NFSP, which cannot be applied at all with continuous actions. Additionally, because PSRO might require an exponential number of pure strategies, NXDO outperforms PSRO on games that require mixing over multiple timesteps. To demonstrate the effectiveness of our approach on these types of games, we run experiments on two sets of environments. The first, $m$-Clone Leduc, is similar to Leduc poker but with every call, fold, and bet action duplicated $m$ times. The second, the Loss Game, is a sequential continuous-action multidimensional optimization game in which agents simultaneously adjust parameters to maximize or minimize a complex loss function. We show that tabular XDO greatly outperforms PSRO, CFR, and XFP (Heinrich et al., 2015) on $m$-Clone Leduc. We also show that NXDO outperforms both PSRO and NFSP on $m$-Clone Leduc and on the continuous-action Loss Game, where NFSP is provided a binned discrete action space.

To summarize, our contributions are as follows:

- We present a tabular extensive-form double oracle algorithm, XDO, that terminates in a linear number of iterations in the number of infostates.
- We present a neural version of XDO, NXDO, that outperforms PSRO and NFSP on both modified Leduc poker and sequential continuous-action games. NXDO is the first method that can find an approximate NE in high-dimensional continuous-action sequential games.

## 2 Background

### 2.1 Extensive-Form Games

We consider partially-observable stochastic games (Hansen et al., 2004) which correspond to perfect-recall extensive-form games (from here on referred to as extensive-form games). An extensive-form

game progresses through a sequence of player actions, and has a **world state** $w \in \mathcal{W}$ at each step. In an $N$-player game, $\mathcal{A} = \mathcal{A}_1 \times \cdots \times \mathcal{A}_N$ is the space of joint actions for the players. $\mathcal{A}_i(w)$ denotes the set of legal actions for player $i \in \mathcal{N} = \{1, \ldots, N\}$ at world state $w$ and $a = (a_1, \ldots, a_N) \in \mathcal{A}$ denotes a joint action. At each world state, after the players choose a joint action, a transition function $\mathcal{T}(w, a) \in \Delta^{\mathcal{W}}$ determines the probability distribution of the next world state $w'$. Upon transition from world state $w$ to $w'$ via joint action $a$, player $i$ makes an **observation** $o_i = \mathcal{O}_i(w, a, w')$. In each world state $w$, player $i$ receives a reward $\mathcal{R}_i(w)$.

A **history** is a sequence of actions and world states, denoted $h = (w^0, a^0, w^1, a^1, \ldots, w^t)$, where $w^0$ is the known initial world state of the game. $\mathcal{R}_i(h)$ and $\mathcal{A}_i(h)$ are, respectively, the reward and set of legal actions for player $i$ in the last world state of a history $h$. An **infostate** for player $i$, denoted by $s_i$, is a sequence of that player's observations and actions up until that time $s_i(h) = (a_i^0, o_i^1, a_i^1, \ldots, o_i^t)$. Define the set of all infostates for player $i$ to be $\mathcal{I}_i$. The set of histories that correspond to an infostate $s_i$ is denoted $\mathcal{H}(s_i) = \{h : s_i(h) = s_i\}$, and it is assumed that they all share the same set of legal actions $\mathcal{A}_i(s_i(h)) = \mathcal{A}_i(h)$.

A player's **policy** $\pi_i$ is a function mapping from an infostate to a probability distribution over actions. A **policy profile** $\pi$ is a tuple $(\pi_1, \ldots, \pi_N)$. All players other than $i$ are denoted $-i$, and their policies are jointly denoted $\pi_{-i}$. A policy for a history $h$ is denoted $\pi_i(h) = \pi_i(s_i(h))$ and $\pi(h)$ is the corresponding policy profile. We also define the transition function $\mathcal{T}(h, a_i, \pi_{-i}) \in \Delta^{\mathcal{W}}$ as a function drawing actions for $-i$ from $\pi_{-i}$ to form $a = (a_i, a_{-i})$ and to then sample the next world state $w'$ from $\mathcal{T}(w, a)$, where $w$ is the last world state in $h$.

The **expected value (EV)** $v_i^\pi(h)$ for player $i$ is the expected sum of future rewards for player $i$ in history $h$, when all players play policy profile $\pi$. The EV for an infostate $s_i$ is denoted $v_i^\pi(s_i)$ and the EV for the entire game is denoted $v_i(\pi)$. A **two-player zero-sum** game has $v_1(\pi) + v_2(\pi) = 0$ for all policy profiles $\pi$. The EV for an action in an infostate is denoted $v_i^\pi(s_i, a_i)$. A **Nash equilibrium (NE)** is a policy profile such that, if all players played their NE policy, no player could achieve higher EV by deviating from it. Formally, $\pi^*$ is a NE if $v_i(\pi^*) = \max_{\pi_i} v_i(\pi_i, \pi_{-i}^*)$ for each player $i$.

The **exploitability** $e(\pi)$ of a policy profile $\pi$ is defined as $e(\pi) = \sum_{i \in \mathcal{N}} \max_{\pi_i'} v_i(\pi_i', \pi_{-i})$. A **best response (BR)** policy $\mathbb{BR}_i(\pi_{-i})$ for player $i$ to a policy $\pi_{-i}$ is a policy that maximally exploits $\pi_{-i}$: $\mathbb{BR}_i(\pi_{-i}) = \arg\max_{\pi_i} v_i(\pi_i, \pi_{-i})$. An **$\epsilon$-best response ($\epsilon$-BR)** policy $\mathbb{BR}_i^\epsilon(\pi_{-i})$ for player $i$ to a policy $\pi_{-i}$ is a policy that is at most $\epsilon$ worse for player $i$ than the best response: $v_i(\mathbb{BR}_i^\epsilon(\pi_{-i}), \pi_{-i}) \geqslant v_i(\mathbb{BR}_i(\pi_{-i}), \pi_{-i}) - \epsilon$. An **$\epsilon$-Nash equilibrium ($\epsilon$-NE)** is a policy profile $\pi$ in which, for each player $i$, $\pi_i$ is an $\epsilon$-BR to $\pi_{-i}$.

A **normal-form game** is a single-step extensive-form game. An extensive-form game induces a normal-form game in which the legal actions for player $i$ are its deterministic policies $\times_{s_i \in \mathcal{I}_i} \mathcal{A}_i(s_i)$. These deterministic policies are called **pure strategies** of the normal-form game. Since each deterministic policy specifies one action at every infostate, there are an exponential number of pure strategies in the number of infostates. A **mixed strategy** is a distribution over a player's pure strategies. Two policies $\pi_i^1$ and $\pi_i^2$ for player $i$ are said to be **realization-equivalent** if for any fixed strategy profile of the other player, both $\pi_i^1$ and $\pi_i^2$, define the same probability distribution over the states of the game.

**Theorem 1** (Kuhn's Theorem (Kuhn & Tucker, 1953)). *Any mixed strategy in the normal form of a game is realization equivalent to a policy in the extensive form of that game, and vice versa.*

## 3 Related Work

There has been much recent work on non-game-theoretic multi-agent RL (Foerster et al., 2018; Lowe et al., 2017; Rashid et al., 2018; Bansal et al., 2017). Most of this work focuses on games with more than two players such as multi-agent cooperative games or mixed competitive-cooperative scenarios. In cooperative environments, self-play has empirically been shown to find an approximate NE (Lowe et al., 2017; Majumdar et al., 2020), but can be brittle when cooperating with agents it hasn't trained with (Lanctot et al., 2017). Self-play reinforcement learning has achieved expert level performance on video games (Vinyals et al., 2019; Berner et al., 2019; Jaderberg et al., 2019), but is not guaranteed to converge to an approximate NE.

Extensive-form fictitious play (XFP) (Heinrich et al., 2015) and counterfactual regret minimization (CFR) (Zinkevich et al., 2008) extend Fictitious Play (FP) (Brown, 1951) and regret matching (Hart &

Mas-Colell, 2000), respectively, to extensive-form games. Deep CFR (Brown et al., 2019; Steinberger, 2019; Li et al., 2018) is a general method that trains a neural network on a buffer of counterfactual values. However, Deep CFR uses external sampling, which may be impractical for games with a large branching factor, such as Stratego and Barrage Stratego. DREAM (Steinberger et al., 2020) and ARMAC (Gruslys et al., 2020) are model-free regret-based deep learning approaches. DREAM and ARMAC have achieved good results in poker games, but since they are based on MCCFR, like Deep CFR, they will not scale to games with continuous actions.

Our work is related to pruning approaches (Brown & Sandholm, 2015a; Brown et al., 2017). These methods start with all actions and sequentially remove actions that have low expected value. XDO instead starts with no actions and sequentially adds actions. Our work is also related to methods that automatically find abstractions (Brown & Sandholm, 2015b; Čermák et al., 2017).

Close to our work, Bosansky et al. (2014) develop a sequence-form double oracle (SDO) algorithm. The SDO algorithm iteratively adds sequence-form BRs to a population and then computes a meta-Nash on a restricted sequence-form game where only sequences in the population are allowed. In contrast, XDO iteratively adds extensive-form BRs to a population and then computes a meta-Nash on a restricted extensive form game where only actions in the population are allowed. DO, SDO, and XDO are fundamentally different because they operate on the normal form, sequence form, and extensive form, respectively. We give a detailed description of the difference between XDO and SDO in the supplementary materials.

### 3.1 Neural Fictitious Self Play (NFSP)

Neural Fictitious Self Play (NFSP) (Heinrich & Silver, 2016) approximates XFP by progressively training a best response against an average of all past policies using reinforcement learning. The average policy is represented by a neural network and is trained via supervised learning using a replay buffer of past best response actions. Each episode, both players either play from their best response policy with probability $\eta = 0.1$ or with their average policy with probability $1 - \eta$. This experience is then added to the best response circular replay buffer and is used to train the best response for both players with off-policy DQN. If a player plays with their best response policy, the data is also added to the average policy reservoir replay buffer and is used to train the average policy via supervised learning.

### 3.2 Policy Space Response Oracles (PSRO)

The Double Oracle algorithm (McMahan et al., 2003) is an algorithm for finding a NE in normal-form games. The algorithm works by keeping a population of policies $\Pi^t$ at time $t$. Each iteration a meta-Nash Equilibrium (meta-NE) $\pi^{*,t}$ is computed for the game restricted to policies in $\Pi^t$. Then, a best response to this meta-NE for each player $\mathbb{BR}_i(\pi^{*,t}_{-i})$ is computed and added to the population $\Pi^{t+1}_i = \Pi^t_i \cup \{\mathbb{BR}_i(\pi^{*,t}_{-i})\}$ for $i \in \{1, 2\}$.

Policy Space Response Oracles (PSRO) (Lanctot et al., 2017) approximates the Double Oracle algorithm. The meta-NE is computed on the empirical game matrix $U^\Pi$, given by having each policy in the population $\Pi$ play each other policy and tracking average utility in a payoff matrix. In each iteration, an approximate best response to the current meta-NE over the policies is computed via any reinforcement learning algorithm. Pipeline PSRO parallelizes PSRO with convergence guarantees (McAleer et al., 2020).

#### 3.2.1 PSRO Hard Instance

A primary issue with PSRO is that it is based on a normal-form algorithm, and the number of pure strategies in a normal-form representation of an extensive-form game is exponential in the number of infostates. In contrast, our approach implements the double oracle algorithm directly in the extensive-form game, overcoming this problem and terminating in a linear number of iterations in the number of infostates. The following example helps illustrate this point.

Consider the game in Figure 1. In this game, first, player 1 chooses which Rock Paper Scissors (RPS) game both players play. After player 1 chooses the RPS game, both players know which RPS game they are playing. Then both players simultaneously play an action in that RPS game. There are 6 pure strategies for player 1, denoted R1, P1, S1, R2, P2, S2. But there are 9 pure strategies for player

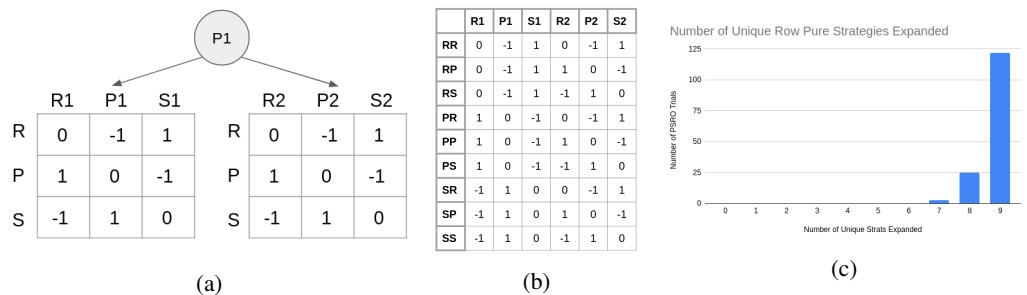

Figure 1: PSRO hard instance. (a) Player 1 first chooses which RPS game both players play. Both players know which RPS game they are playing. Then both players simultaneously make their move. (b) The normal form game. Player 2 has 9 pure strategies. (c) The proportion of PSRO trials that expanded each possible number of pure strategies for player 2. In the majority of trials, PSRO had to expand all possible pure strategies.

2. A pure strategy for player 2 specifies what move they play at each infostate. If player 2 played Rock in the first infostate and Paper in the second, that pure strategy is denoted RP. Note that if we generalize this game by including more RPS games and more actions in each game, the number of pure strategies for player 2 will be $|A|^{|\mathcal{I}|}$, where $|A|$ is the number of actions and $|\mathcal{I}|$ is the number of RPS games.

We conduct an experiment where we run PSRO with oracle BRs on this game with a random starting population each time. We find that PSRO expands all 9 row (player 2) pure strategies the majority of the time, expanding all 9 strategies in 122 out of 150 trials. These results are shown in Figure 1c. We also find that the column player (player 1) expands all 6 pure strategies in all 150 trials.

# 4 Extensive-Form Double Oracle (XDO)

We propose Extensive-Form Double Oracle (XDO), a double-oracle (DO) algorithm designed for two-player zero-sum extensive-form games (Algorithm 1). As in other DO algorithms, XDO maintains a population of pure strategies, and in each iteration computes a meta-NE of this population. Then the algorithm finds a best response (BR) to the meta-NE and adds it to the population.

In XDO, the population induces a different restricted game, and therefore a different population meta-NE, than in PSRO (Lanctot et al., 2017). In PSRO, a restricted normal-form game is induced by the empirical payoff matrix of population strategies. In XDO, a restricted extensive-form game is induced through a transformation on the original base extensive-form game that restricts the allowed actions at each infostate to only those suggested by any strategy in the population.

XDO uses a tabular method such as CFR (Zinkevich et al., 2008) or XFP (Heinrich et al., 2015) to solve the restricted game. The algorithm terminates after an iteration in which neither of the players finds a BR that outperforms the meta-NE. When this happens, the meta-NE policies are approximate BRs to each other in the original game as well, and the meta-NE is therefore an approximate NE of the original game.

Importantly, at each but the final iteration of XDO, at least one player adds some new action at some non-terminal infostate, because a BR cannot outperform the meta-NE with only restricted-game actions. The number of iterations that XDO takes to terminate is therefore at most the number of infostates, including terminal ones. In contrast, the best known guarantee for the number of iterations that PSRO takes to terminate (Proposition 2) is exponential in the number of infostates, because PSRO may need to add all pure strategies to the population. Moreover, computing the meta-NE in PSRO may become intractable in later iterations as the population size increases, while in XDO it is bounded by the unrestricted game.

Formally, XDO keeps a population of pure strategies $\Pi^t$ at time $t$. Each iteration, a restricted extensive-form game is created and a NE to the restricted game is computed. The restricted game is created by taking the original game and restricting the actions at every infostate $s_i$ to be only the

---

**Algorithm 1** XDO

---

1: Input: initial population $\Pi^0$
2: **repeat**
3:    Define restricted game for $\Pi^t$ via eq. (1)
4:    Get $\epsilon$-NE policy $\pi^{r*}$ of restricted game
5:    Find $\mathbb{BR}_i(\pi^{r*}_{-i})$ for $i \in \{1, 2\}$
6:    **if** $v_i(\mathbb{BR}_i(\pi^{r*}_{-i}), \pi^{r*}_{-i}) \leqslant v_i(\pi^{r*}) + \epsilon$ for both $i$ **then**
7:       Terminate
8:    $\Pi^{t+1}_i = \Pi^t_i \cup \mathbb{BR}_i(\pi^{r*}_{-i})$ for $i \in \{1, 2\}$

---

actions where there exists a policy in the population $\Pi^t$ that chooses that action at that infostate:

$$\mathcal{A}^r_i(s_i) = \{a \in \mathcal{A}_i(s_i) : \exists \pi_i \in \Pi^t_i \text{ s.t. } \pi_i(s_i, a) = 1\}. \tag{1}$$

An $\epsilon$-NE policy $\pi^{r*}$ is then computed in this restricted game via a tabular method such as CFR and is extended to the full game by defining arbitrary actions on infostates not encountered in the restricted game. Next, BRs to this restricted game meta-NE $\mathbb{BR}_1(\pi^{r*}_2)$ and $\mathbb{BR}_2(\pi^{r*}_1)$ are computed via an oracle. These BRs are then added to the population of policies: $\Pi^{t+1}_i = \Pi^t_i \cup \mathbb{BR}_i(\pi^{r*}_{-i})$ for $i \in \{1, 2\}$.

The algorithm terminates when neither player benefits more than $\epsilon$ from deviating from the meta-NE to the BR, indicating that the meta-NE is an $\epsilon$-NE also in the original game (Proposition 1).

To illustrate how XDO works, we demonstrate a simple game in Figure 2. The algorithm starts with empty populations. At the first iteration (left diagram), player 1 adds a BR that plays Left at the first infostate (the root) and Right at the second one. Player 2 simultaneously adds a BR that plays Right at their single infostate. The restricted game now consists of only these added actions. At the second iteration (middle diagram), player 1 adds a BR that plays Right at both infostates, and player 2's BR still plays Right. The restricted game now includes both actions for the root infostate, but only Right is in the meta-NE. Next, in the third iteration (right diagram), player 1 keeps the same BR, while player 2's BR plays Left. In the meta-NE of this final restricted game, player 1 plays Left and Right with equal probability at the first infostate, and player 2 plays Left with probability 0.37 and Right with probability 0.63. Since the BRs to this meta-NE do not add any new actions, XDO terminates, and the meta-NE is the NE for the full game. Note that in this example, most actions are needed to find a NE. In games like this, it would be faster to simply solve the original game from the beginning. However, certain games, such as those in our experiments, have Nash equilibria that only need to mix over a small subset of actions (Schmid et al., 2014), in which case solving the XDO restricted game will be much faster than solving the original game.

### 4.1 Theoretical Considerations

In this section, we present a theoretical analysis of XDO and compare it with PSRO. Our first proposition states that, when XDO terminates, the final meta-NE of the restricted game is an approximate NE of the full game.

**Proposition 1.** *In XDO with an $\epsilon_1$-BR oracle, let $\pi^{r*}$ be the final $\epsilon_2$-NE in the restricted game. Then $\pi^{r*}$ is an $(\epsilon_1 + \epsilon_2)$-NE in the full game.*

*Proof.* Due to limited space, all proofs are contained in the supplementary materials. □

The next two propositions show an exponential gap in the known guarantees for the number of iterations in which PSRO and XDO terminate. If each non-terminal infostate allows $A$ different actions, PSRO is guaranteed to terminate in $\sum_i A^{|\bar{\mathcal{I}}_i|}$ iterations, where $\bar{\mathcal{I}}_i$ is the set of non-terminal infostates for player $i$, while XDO is guaranteed to terminate in $\sum_i |\mathcal{I}_i|$ iterations.

**Proposition 2.** *Normal-form DO terminates in at most $\sum_i \prod_{s_i \in \mathcal{I}_i} |\mathcal{A}_i(s_i)|$ iterations.*

**Proposition 3.** *XDO terminates in at most $\sum_i |\mathcal{I}_i|$ iterations.*

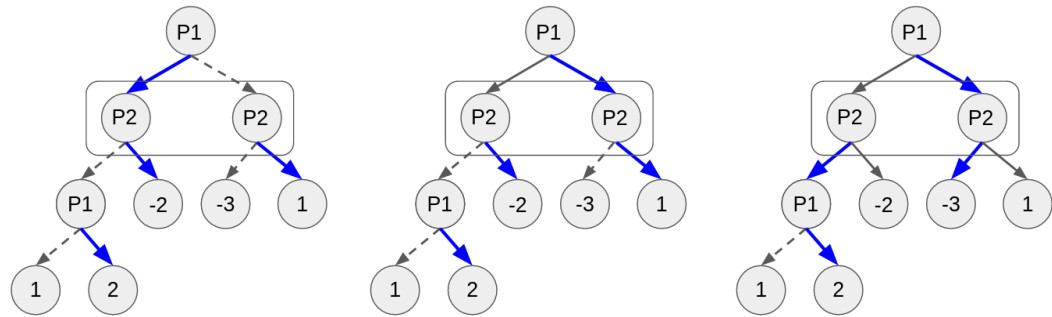

Figure 2: Three iterations of XDO (left to right). In these extensive-form game diagrams, player 1 (P1) plays at the root, then P2 plays without knowing P1's action, and if both played Left P1 plays another action. P1's reward is number at the reached leaf. Actions in the restricted game are solid, vs. dashed outside the restricted game. Meta-NE actions are blue, vs. black not in the meta-NE.

**Tightness of the guarantees.** The guarantees in Proposition 2 and Proposition 3 are tight in the sense that they are achieved in some games, but more nuanced analysis is required to identify easier cases where these bounds overestimate the complexity of the algorithms. Both PSRO and XDO often outperform these guarantees and terminate in fewer iterations. A case in which PSRO expands all pure normal-form strategies of an extensive-form game is described in the supplementary materials.

**XDO can add multiple actions in each iteration.** In practice, XDO often outperforms the guarantee of Proposition 3 because it adds multiple actions in each iteration. Here we present and analyze a family of games in which XDO terminates in asymptotically fewer iterations than suggested by the bound in Proposition 3.

In a generalized matching pennies (GMP) game, both players simultaneously choose one of $n$ actions. The payoff to player 1 is $n-1$ if the actions match, or $-1$ if they are different. In a $k$-GMP game (Figure 3), a chance node first selects an index $j$ between 1 and $k$, and then the players play the $j$'th of $k$ identical GMP games. The following proposition provides a tighter performance bound for XDO in this case, $2n$ iterations instead of $\sum_i |\mathcal{I}_i| = 2k(n+1)$ (there are $kn$ terminal infostates for each player). For PSRO on $k$-GMP, no tighter bound than the $2n^k$ indicated by Proposition 2 is known.

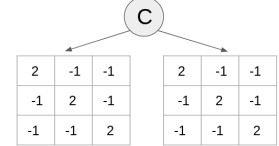

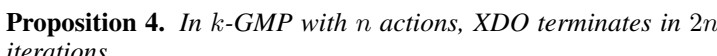

Figure 3: A 2-GMP game with $n = 3$ actions. The chance node selects uniformly at random which generalized matching pennies game is played. Both players know which stage game they play.

**Proposition 4.** *In $k$-GMP with $n$ actions, XDO terminates in $2n$ iterations.*

**Size of the restricted game.** The number of iterations in each algorithm does not provide the full picture of their performance, since iterations can require vastly different computation times. Intuitively, the restricted game in XDO is much larger than in PSRO when both algorithms have the same population size, because XDO induces an extensive-form restricted game with all discovered actions, while PSRO induces a normal-form restricted game with population policies as actions. However, as both algorithms progress, the XDO restricted game is bounded in size by the original game, while PSRO can induce a game with exponentially many actions.

**XDO for sparse-support policies.** XDO is useful when the policies in the population do not cover the full original game, because when they do, finding the restricted game meta-NE is as hard as solving the original game. The motivation behind XDO is that, in games where the NE policies are supported by few actions in most infostates, XDO has the potential to quickly find these actions and terminate without expanding the full game.

To analyze this behavior, consider the $m$-clone GMP game, in which there are $mn$ actions partitioned into $n$ equal classes. The actions of the two players are considered a match (with payoff $n-1$ to player 1) if they belong to the same class. In $(k, m)$-clone GMP, a chance node selects among $k$ identical $m$-clone GMP games. The following proposition states that in $(k, m)$-clone GMP with $n$

**Algorithm 2** NXDO

1: Input: initial population $\Pi^0$
2: **repeat**
3:     Define restricted game for $\Pi^t$ via eq. (2)
4:     Get $\epsilon$-NE policy $\pi^{r*}$ of restricted game via NFSP
5:     Find $\mathbb{BR}_i(\pi_{-i}^{r*})$ for $i \in \{1, 2\}$ via DRL
6:     $\Pi_i^{t+1} = \Pi_i^t \cup \mathbb{BR}_i(\pi_{-i}^{r*})$ for $i \in \{1, 2\}$

classes, XDO terminates after adding at most $2n$ actions for each player, instead of the full game of $kmn$ actions.

**Proposition 5.** *In $(k, m)$-clone GMP with $n$ classes, XDO adds at most $2n$ actions for each player.*

**PSRO lower bound.**    Similarly to XDO, PSRO can also outperform the guarantee of Proposition 2 in certain cases. Generically, however, the linear upper bound on XDO established by Proposition 3, $\sum_i |\mathcal{I}_i|$, is also a *lower bound* on the normal-form population size of pure strategies that is needed to support a NE in PSRO. To show this, consider a perturbed $k$-GMP game, in which the payoffs in each GMP game are slightly modified to induce $k$ distinct NE. The following proposition establishes a linear lower bound for PSRO in perturbed $k$-GMP games.

**Proposition 6.** *There exist perturbed $k$-GMP games with $n$ actions in which PSRO cannot terminate in fewer than $k(n - 1) + 1$ iterations.*

## 5   Neural Extensive-Form Double Oracle (NXDO)

Neural Extensive-Form Double Oracle (NXDO) extends XDO to large games through deep reinforcement learning (DRL). Instead of using an oracle best response, NXDO instead uses approximate best responses that are trained via any DRL algorithm, such as PPO (Schulman et al., 2017) or DDQN (Van Hasselt et al., 2016). Instead of representing the restricted game explicitly as the set of allowed actions in every infostate, to create its restricted game, NXDO replaces the original game action space with a discrete set of meta-actions, each corresponding to a population policy to which the actual action choice is delegated.

Formally, NXDO (Algorithm 2) keeps a population of DRL policies $\Pi^t$ at time $t$. Each iteration, a restricted extensive-form game is created and a meta-NE to the restricted game is computed. The restricted game has meta-actions at every infostate that pick one policy from the population

$$\forall s_i \in \mathcal{I}_i \quad \mathcal{A}_i^r(s_i) = \{1, 2, ..., |\Pi_i^t|\}. \tag{2}$$

While the action space differs, the restricted game states, observations, and histories remain the same as in the original game. After each player selects a meta-action that indicates a population policy, an action is sampled from that population policy and used for the world state transition. With $\pi_i^1, \ldots, \pi_i^{|\Pi_i|}$ the population policies for player $i$, the transition function in the restricted game satisfies

$$\mathcal{T}^r(h, a^r, w') = \sum_a \prod_i \pi_i^{a_i^r}(s_i(h), a_i)\mathcal{T}(h, a, w'). \tag{3}$$

With the restricted game thus defined, an $\epsilon$-meta-NE $\pi^{r*}$ is computed in this restricted game via a DRL method for finding NE, such as NFSP (Heinrich & Silver, 2016) or DREAM (Steinberger et al., 2020). Approximate BRs $\mathbb{BR}_1(\pi_2^{r*})$ and $\mathbb{BR}_2(\pi_1^{r*})$ to this meta-NE are computed via a DRL algorithm, such as PPO or DDQN. These BRs are then added to the population of policies: $\Pi_i^{t+1} = \Pi_i^t \cup \mathbb{BR}_i(\pi_{-i}^{r*})$ for $i \in \{1, 2\}$. Provided that the DRL best responses are sufficiently close to oracle best responses and the inner-loop solver finds a sufficiently close approximate NE of the restricted game, NXDO inherits the same convergence properties as XDO. In practice, contemporary DRL methods lack any guarantee of providing approximate NE or BRs. Nevertheless, we show experimentally that exploitability can decrease through execution of NXDO faster than it does for PSRO and NFSP.

Because the original game action space has no influence on the NXDO restricted game action space, NXDO, like PSRO, is compatible with extremely large and continuous action spaces, provided

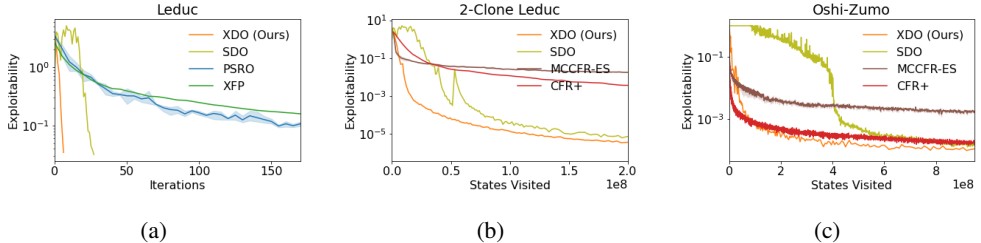

(a)                              (b)                              (c)

Figure 4: (a) Exploitability in Leduc poker of XDO vs. PSRO, SDO, and XFP with oracle BRs throughout their iterations; (b) Exploitability in 2-Clone Leduc poker as a function of the number of game states visited by XDO, SDO, CFR$^+$, and MCCFR with external sampling (ES); (c) Exploitability in Oshi-Zumo as a function of the number of game states visited by XDO, SDO, CFR$^+$, and MCCFR-ES

that the deep RL BRs can operate in such an action space well. We demonstrate this capability in our Loss Game experiments, in which NXDO and PSRO use continuous-action PPO BRs. While no convergence guarantees are known in continuous-action games, NXDO and PSRO empirically produce meta-Nash strategies that are hard to exploit in our experiments.

A drawback of meta-actions that delegate actions to population policies is that, as in PSRO, the number of meta-actions grows linearly with the number of iterations. This can eventually make the restricted game harder to solve than the original game. In our experiments, however, NXDO achieves significant improvements in exploitability within a very small number of iterations, such that the issue of action delegation does not become an obstacle. In discrete action space games where it is tractable, we also consider a variant, NXDO-VA, where the restricted game is explicitly calculated and defined with valid and invalid original-game actions in the same way as with tabular XDO, using equation (1). Because its restricted game action space size is at most equal to that of the original game, NXDO-VA does not suffer from the aforementioned drawback.

## 6  Experiments

For the tabular experiments, we use XDO with an oracle best response (BR) and CFR$^+$ (Tammelin, 2014) for the inner-loop meta-NE solver. We compare XDO with PSRO (Lanctot et al., 2017) and XFP (Heinrich et al., 2015), which use oracle BRs as well. We also compare with CFR$^+$, and for both CFR$^+$ and XFP we follow the implementations in OpenSpiel (Lanctot et al., 2019). We compare to SDO, using the same oracle BR and meta-NE solver as XDO. Since CFR$^+$, XFP, SDO, and XDO are deterministic, we do not plot error bars for these algorithms. For the neural experiments, we use NXDO with NFSP (Heinrich & Silver, 2016) as the meta-NE solver and PSRO with FP (Brown, 1951) as the meta-NE solver. NXDO and PSRO share the same BR configuration, using DDQN (Van Hasselt et al., 2016) for discrete action spaces and PPO (Schulman et al., 2017) for continuous action spaces. We compare these algorithms on $m$-Clone Leduc poker, Oshi-Zumo, and the Loss Game, described in the supplementary materials.

### 6.1  Tabular Experiments with XDO

Finding a normal-form meta-NE can be much less efficient and more exploitable than finding an extensive-form meta-NE. This means that PSRO will often require many more pure strategies to achieve a similar level of exploitability to XDO. Figure 4a summarizes the results of running XDO, SDO, XFP, and PSRO with an oracle BR on Leduc poker. Even after 150 iterations, PSRO remains significantly more exploitable than XDO is at 7 iterations. XDO achieves exploitability of 0.1 in over 20x fewer iterations than PSRO. In large games where calculating many approximate BRs via reinforcement learning is expensive, requiring vastly more iterations can render PSRO infeasible. XFP performs similarly to PSRO in Leduc, as its average policy is equivalent to uniformly mixing population strategies at the root of the game. SDO takes more iterations than XDO to achieve low exploitability, because it adds fewer actions to the restricted game per iteration.

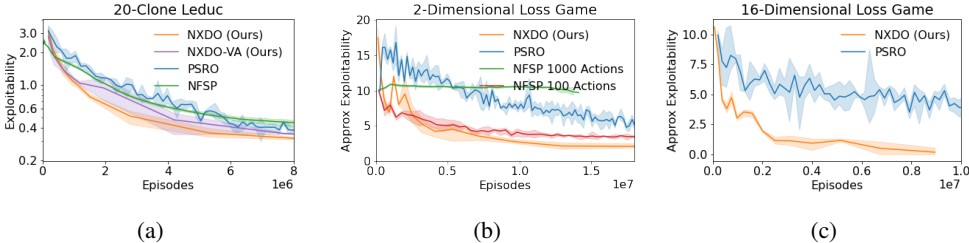

(a)                              (b)                              (c)

Figure 5: (a) Exploitability in 20-Clone Leduc poker of NXDO, NXDO-VA, PSRO, and NFSP as a function of episodes gathered; (b and c) Approximate exploitability as a function of episodes gathered in the continuous-action Loss Game of NXDO, PSRO, and NFSP (with binned discrete actions).

We compare XDO with SDO, CFR$^+$, and MCCFR (Lanctot et al., 2009) in 2-Clone Leduc poker. In Figure 4b, we plot the exploitability of these algorithms as a function of the number of game states visited by the algorithms. Since XFP and PSRO only use BR oracles, we do not include them in this analysis. Since CFR$^+$ updates every infostate every iteration, as we increase the number of cloned actions, the performance of CFR$^+$ will deteriorate. In contrast, XDO will tend to not add cloned actions, which allows the inner-loop CFR$^+$ to expand fewer infostates. These results for XDO are with a deterministic best response oracle. We found that if XDO randomly chose a best response instead, then XDO would still outperform CFR$^+$, but not by as much. The results in Figure 4c suggest that on Oshi-Zumo XDO outperforms SDO, MCCFR, and, to a lesser degree, CFR$^+$. We do not include XDO with MCCFR as the inner loop solver because it did not perform as well as XDO with CFR$^+$.

## 6.2 Neural Experiments with NXDO

In Figure 5a, we compare the exploitability of NXDO and NXDO-VA with PSRO and NFSP on 20-Clone Leduc poker. DDQN is used to train NXDO and PSRO BRs. Similar to the tabular experiments, we find that NXDO outperforms both methods. However, we find that the margin by which NXDO outperforms these methods is smaller than in tabular experiments. This can be attributed to the large proportion of experience required by NFSP to solve the restricted game relative to experience spent learning BRs. Training details and an analysis on the proportion of experience spent in the inner vs outer loop of NXDO are included in supplementary materials.

We also test NXDO and PSRO on the 2D and 16D continuous-action Loss Game, shown in Figures 5b and 5c respectively. PPO is used to train NXDO and PSRO BRs. In the 2D action space game, we compare to NFSP with a binned discrete action space. Because calculating exact exploitability is intractable for the Loss Game, for each algorithm checkpoint, we measure approximate exploitability by training a continuous-action PPO best response against it until saturation and measuring the BR's final mean reward. NXDO outperforms NFSP in the 2D game while operating in a continuous action space and outperforms PSRO in the 2D and 16D game which we conjecture is due to more effective use of population strategies in its restricted game.

# 7 Conclusion

PSRO and NXDO are the only existing game-theoretic deep RL methods that can work on large continuous-action games. In games where the NE must mix in many infostates, but only a small fraction of all actions are in the support of the NE at each infostate, we expect XDO and NXDO to outperform PSRO, because PSRO may require a superlinear, or even exponential number of pure strategies. We also expect XDO and NXDO to outperform CFR and NFSP, respectively, on discrete-action games where the NE only needs to mix over a small fraction of actions. This is because CFR and NFSP scale poorly with the number of actions in the game, but XDO and NXDO tend to discover a set of relevant actions and ignore actions that are dominated or redundant. We hypothesize that games with these properties are prevalent across a number of domains such as large board games, video games, and robotics applications.

## 8 Acknowledgements

The authors would like to thank Marc Lanctot and Julien Perolat for interesting and helpful discussions, and Chinmay Tyagi for help in prototyping related concepts.

Roy Fox's research is partly funded by the Hasso Plattner Foundation.

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
