# Supplementary Materials For XDO: A Double Oracle Algorithm for Extensive-Form Games

## 1 Proofs

**Proposition 1.** *In XDO with an $\epsilon_1$-BR oracle, let $\pi^{r*}$ be the final $\epsilon_2$-NE in the restricted game. Then $\pi^{r*}$ is an $(\epsilon_1 + \epsilon_2)$-NE in the full game.*

*Proof.* For each $i \in \{1, 2\}$, let $\mathbb{BR}_i^{\epsilon_1}(\pi_{-i}^{r*})$ be player $i$'s $\epsilon_1$-BR to $\pi_{-i}^{r*}$ obtained in the last iteration. By the termination condition

$$v_i(\pi^{r*}) \geqslant v_i(\mathbb{BR}_i^{\epsilon_1}(\pi_{-i}^{r*}), \pi_{-i}^{r*}) - \epsilon_2 \tag{1}$$

$$\geqslant \max_{\pi_i'} v_i(\pi_i', \pi_{-i}^{r*}) - \epsilon_1 - \epsilon_2, \tag{2}$$

where the last inequality follows from $\mathbb{BR}_i^{\epsilon_1}(\pi_{-i}^{r*})$ being an $\epsilon_1$-best response to $\pi_{-i}^{r*}$. □

**Proposition 2.** *Normal-form DO terminates in $\sum_i \prod_{s_i \in \mathcal{I}_i} |\mathcal{A}_i(s_i)|$ iterations.*

*Proof.* In each iteration of DO, at least one player adds a new normal-form pure strategy to the population. The space of pure strategies for player $i$ has size $\prod_{s_i \in \mathcal{I}_i} |\mathcal{A}_i(s_i)|$, because each normal-form pure strategy specifies an action at each infostate for that player. □

**Proposition 3.** *XDO terminates in $\sum_i |\mathcal{I}_i|$ iterations.*

*Proof.* Consider an infostate $s_i' = (a_i^0, o_i^1, \ldots, a_i^t, o_i^{t+1})$ for player $i$ as *covered* in the restricted game if any of player $i$'s population policies chooses action $a_i^t$ in infostate $s_i = (a_i^0, o_i^1, \ldots, a_i^{t-1}, o_i^t)$. At each but the final iteration, at least one player $i$ has $v_i(\mathbb{BR}_i(\pi_{-i}^{r*}), \pi_{-i}^{r*}) > v_i(\pi^{r*}) + \epsilon$. Since $\pi_i^{r*}$ is an $\epsilon$-BR to $\pi_{-i}^{r*}$ in the restricted game, the BR $\mathbb{BR}_i(\pi_{-i}^{r*})$ must be choosing at least some action $a_i$ at some non-terminal infostate $s_i$ that was not previously chosen by any population policy. Adding this action to the restricted game covers at least one previously uncovered infostate: all infostates $s_i' = (s_i, a_i, o_i)$, for any observation $o_i$. All infostates will therefore be covered in at most $\sum_i |\mathcal{I}_i|$ iterations, at which point the next iteration must terminate. □

**Proposition 4.** *In $k$-GMP with $n$ actions, XDO terminates in $2n$ iterations.*

*Proof.* In a given iteration, consider the restricted game for a single GMP game. If player 2 is allowed an action that player 1 is not, such an action will be player 2's NE, and player 1's BR will add that action. If player 2 is not allowed an action unavailable to player 1, player 2's BR will be a new action unavailable to player 1, if one exists. Thus at least one of the players add a new action in every GMP game in parallel, until both players add all actions. □

**Proposition 5.** *In $(k, m)$-clone GMP with $n$ classes, XDO adds at most $2n$ actions for each player.*

*Proof.* The proof repeats that of Proposition 4, but considering classes instead of actions, because it does not matter which member of a class is added. Once at least one member of each class is added to the restricted game, the meta-NE has full-game exploitability 0, and XDO terminates. In iterations

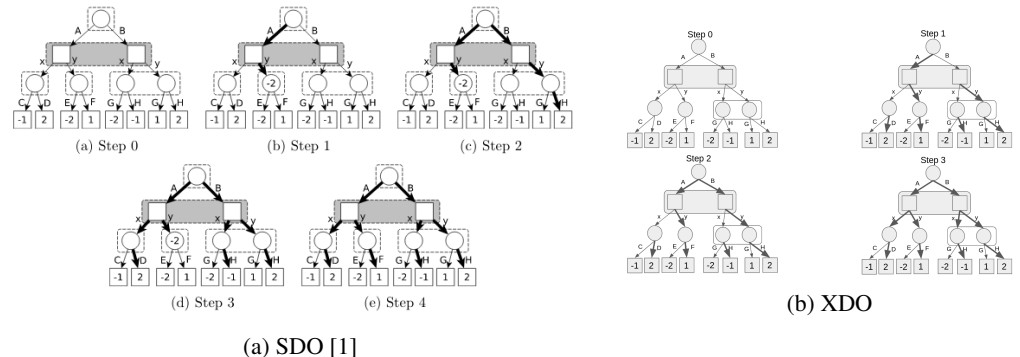

(a) SDO [1]

(a) Step 0 (b) Step 1 (c) Step 2 (d) Step 3 (e) Step 4

(b) XDO

Figure 1

where a BR for a player does not add a new class, it may add a new action member of an existing class. In total, $2n$ actions may be added for each player. □

**Proposition 6.** *There exist perturbed $k$-GMP games with $n$ actions in which PSRO cannot terminate in fewer than $k(n-1)+1$ iterations.*

*Proof.* For each policy $\pi_2 \in (\Delta(n))^k$ for player 2, consider the perturbed $k$-GMP game that gives player 1 payoff $\frac{1}{\pi_2(j,a)}$ for matching action $a$ in stage game $j$. In the NE for this game, player 2 has policy $\pi_2$. This implies that the set of policies that are NE of any perturbed $k$-GMP game has positive $k(n-1)$-dimensional volume.

Consider the space of stochastic policies that can be spanned by mixing a specific population of at most $k(n-1)$ pure strategies. The dimension of this space is at most $k(n-1)-1$. When we consider the union of all such spaces for the finitely many possible populations of this size, this set has zero $k(n-1)$-dimensional volume.

It follows that there exists a perturbed $k$-GMP game $G$, and a neighborhood around player 2's policy in the NE of $G$, such that no policy in that neighborhood is spanned by any population of $k(n-1)$ pure strategies. For sufficiently small $\epsilon$, no $\epsilon$-NE for $G$ can be found until PSRO adds at least $k(n-1)+1$ pure strategies to its population. □

## 2   Comparison to SDO

To illustrate the difference between the two algorithms, we provide an example run of XDO (Figure 1b) and SDO (Figure 1a) on the same game that is presented as an example in section 4.1.3 in the SDO paper [1].

Like in that work, we represent actions that are in the restricted game by bold arrows. This example demonstrates how SDO creates smaller restricted games than XDO because it only considers infostates that can be reached by compatible sequences. In tabular games, SDO results in a cautious approach that only considers a small subset of infostates in order to prevent adding suboptimal actions to the restricted game. However, as we describe below, this will cause obstacles when trying to scale SDO to large games.

Extensive-form pure strategies specify an action at every infostate. In this example we will refer to extensive-form pure strategies by concatenating the actions the strategy takes in every infostate. For example, the pure strategy for the circle player in step 1 in our diagram corresponds to ADFH. Sequence-form pure strategies specify a sequence of actions that must be internally consistent. For example, $\{\varnothing, A, AD\}$ is a valid sequence-form pure strategy but $\{\varnothing, B, AD\}$ is not.

In step 0, both SDO and XDO start with an empty game tree. Let's assume that the default strategy for both algorithms is uniform random.

In step 1, both SDO and XDO add the same best responses to the default strategy for both players. However, SDO adds actual sequences of actions, in particular $\{\varnothing, A, AD\}$ for the circle player and

$\{\emptyset, y\}$ for the box player. Since the AD sequence of actions for the circle player is not compatible with the y action for the square player, the restricted game is the game with A as the available action for circle and y as the available action for square. But since neither AE nor AF are in the sequence population, the restricted game in SDO at this step terminates in a temporary leaf at that point.

In contrast, XDO adds full extensive form strategies, in this case adding ADFH for the circle player and y for the square player. Now the restricted game for XDO is as in step 1 in the diagram. Since XDO adds full extensive form strategies, there is no need to create a temporary leaf node.

After only one iteration, SDO and XDO result in a different restricted game. This is because SDO adds sequences of actions (best responses in sequence form) to the population while XDO adds a pure strategy defined at every infostate (best responses in extensive form). The SDO restricted game is much smaller than the XDO restricted game because SDO only considers infostates that can be reached by compatible sequences in the population. There are three more steps for SDO, which are described in their paper. XDO has only two more steps, which are described in the figure.

There currently exists one main algorithm that can scale up the double oracle approach to large games through deep reinforcement learning, which is PSRO. We propose another, Neural Extensive-Form Double Oracle (NXDO). Similarly extending SDO to large games via neural networks is one possible direction for future work.

# 3  Empirical Tests on Perturbed $k$-GMP

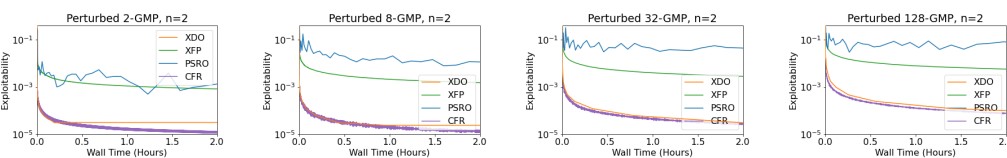

Figure 2: Exploitability vs wall time hours with XDO, XFP, PSRO, and CFR in Perturbed $k$-GMP as the number of subgames rises. XDO and CFR scale well even when there are many subgames while PSRO becomes unable to reach a low exploitability.

We run XDO, XFP, PSRO (DO) and CFR on the Perturbed $k$-GMP games. In perturbed $k$-GMP, a chance node randomly selects one of the $k$ stage games and the outcome is observed by both players. A static perturbation uniformly sampled between $(-1.0, 1.0)$ is added to every GMP payoff where both players select the same action to create a unique equilibrium solution for each of the $k$ subgames. We see that as $k$ is increased from 2 to 128, XDO and CFR exploitability increases but remains relatively low, but PSRO performance relative to computation time dramatically decreases. For this experiment, we use CFR as the meta-NE solver for XDO.

# 4  Game Descriptions

**$m$-Clone Leduc poker:**  $m$-Clone Leduc poker is similar to Leduc poker but with every action duplicated $m$ times, such that instead of a single call, fold, and bet action there are $m$ identical call, fold, and bet actions. As the number of cloned actions increases, we expect the performance of methods based on CFR and FP such as DREAM and NFSP to deteriorate, while the performance of XDO and PSRO remains largely unchanged because the size of the restricted games scale with the size of the meta-game population rather than the size of the action space.

**Oshi-Zumo**  Oshi-Zumo is a zero-sum two-player sequential game where both players try to move a token to the other player's side of the board. Both players start with an allotment of coins and each step simultaneously bet an amount of coins. The player who bets the higher amount gets to move the token toward the opponent. A player wins if they are able to move the token off the board on their opponent's side or if the token is on the opponent's side when either no more coins remain or the time horizon is reached. Reward is 1 for wining, -1 for losing, and 0 for a tie, which can occur when

the token is in the middle of the board when the game ends. We consider a variant with 4 coins, and a board of 3 spaces with a time horizon of 6.

**Loss Game:** The Loss Game is a zero-sum two-player sequential continuous action game in which agents simultaneously apply bounded adjustments to a real valued vector of parameters $\boldsymbol{x} = [x_1, \ldots, x_d]$ in order to optimize a fixed loss function's scalar output $L : \mathbb{R}^d \mapsto \mathbb{R}$ . One agent aims to maximize this output while to other aims to minimize it. Each timestep, agents observe the current vector of parameter values along with the function's scalar output value and provide a bounded continuous action describing an adjustment vector to add to the current parameters. The sum of both agents' adjustments is applied and player 1 is rewarded with the value of the function's output while player 2 is provided with the negative of this value. The game lasts 10 steps and the parameters' adjusted values are preserved after each step.

We consider a 2D action space variation with the loss function $L(x_1, x_2) = \sum_j \sin(x_j)$ and a 16D action space variation with $L(x_1, \ldots, x_{16}) = \sin(\sum_j x_j) + \sum_j \sin(x_j)/16$. These functions were chosen to demonstrate games in which it is advantageous to mix between multiple strategies in many infostates. The 2 and 16 dimensional action spaces where chosen to represent continuous spaces that, respectively, can and cannot be directly binned into a tractable amount of discrete actions. We test the binned discrete action version of the 2D continuous action space Loss Game with both 10 and 100 bins in each of the two dimensions , totalling 100 and 10,000 actions respectively.

# 5 Tabular Experiment Details

For calculating tabular BRs, we use an oracle implementation supplied by the OpenSpiel framework [3].

For tabular PSRO, we use linear programming to solve each meta-NE. To calculate the empirical payoff matrix, we play 100 games per policy combination.

For CFR$^+$, CFR, MCCFR, and XFP, we use the default implementations and settings provided by OpenSpiel.

For tabular XDO, we repeatedly improve the $\epsilon$-NE (e.g. perform CFR iterations) for the restricted game until both of the following conditions are met: *a)* the exploitability of the meta-NE in the restricted game is less than $\epsilon$, and *b)* the exploitability of the meta-NE in the restricted game is less than the exploitability of the meta-NE in the full game. We initialize $\epsilon$ to be $0.35$, and each XDO iteration, we set $\epsilon$ to $0.98 * \epsilon$.

This has two benefits: First, we guarantee that all BRs added to the population are useful, in that they contain at least some out-of-restricted-game action, and thus expand the restricted game. Second, if the restricted game already contains the actions necessary for a NE in the full game, we simply continue to improve the meta-NE ad infinitum, rather than add a new BR and restart the meta-NE solver.

For SDO, we use CFR+ as an inner-loop solver, instead of a linear program as used in the SDO paper [1]. The default pure strategy we use is to choose the first possible action. To find best-response sequences, we compute a tabular BR in the full game.

Unless otherwise stated, all non-deterministic tabular experiments were run with 3 seeds each.

# 6 Neural Experiment Details

For reinforcement learning neural experiments, we use DDQN for discrete action spaces and PPO for the Loss Game continuous action space. The NFSP RL policy is also trained with DDQN. All neural experiments were run with 3 seeds each.

## 6.1 Best Response Stopping Conditions

In both 20-Clone Leduc Poker and the Loss Game, BRs in PSRO and NXDO are trained for a minimum of 4e4 episodes and a maximum of 1e5 episodes. Every 2e4 episodes, a check is performed as to whether the average episode reward is at least 0.01 higher than it was in the last check. If

this check fails, the BR is considered plateaued and is stopped early. This stopping condition was chosen to ensure that BRs are allowed to saturate in performance without spending a redundantly large amount of episodes training.

## 6.2 Meta-NE Solver Stopping Conditions

For NXDO, we use NFSP as the meta-NE solver. Similar to tabular XDO, we increase the amount of experience allocated to solve the meta-NE over multiple NXDO iterations. This is done because adding additional diverse BR policies to the population in early iterations of NXDO is more effective than spending a large (or any) amount of time solving for the restricted game meta-NE with a very small population. In addition to a schedule used to increase the amount of training the for meta-NE solver, for the first $n$ NXDO iterations, we execute a "warm-start" in which we use an untrained NFSP average policy network in lieu of training the meta-NE. After $n$ warm start iterations, we train NFSP to solve for the meta-NE with a schedule that gradually increases the amount of experience NFSP is allocated for training each NXDO iteration. As the NXDO population grows larger, it is advantageous to train the meta-NE solver for longer because the restricted game solution will better approximate the original game NE. We use two different meta-NE stopping condition schedules for 20-Clone Poker and the Loss Game, described below:

The meta-NE stopping condition schedule used for NXDO on 20-Clone Poker solves for an $\epsilon$-NE of the restricted game, where $\epsilon$ is decreased over NXDO iterations. After 7 warm start iterations, $\epsilon$ is initialized at 1.0, and, each NXDO iteration, $\epsilon$ is halved if the final average reward of the previous iteration's BRs is below $(\epsilon + 0.05)$. $\epsilon$ is not allowed to fall below 0.05. The exact stopping condition for NFSP in the inner loop of NXDO is to train for at least 2e5 episodes and then stop when the mean DDQN BR reward against average policies falls below $\epsilon$.

A simpler schedule is used for NXDO on the Loss Game. After 5 warm start iterations, we train NFSP on the restricted game for 1e6 time steps. In each subsequent NXDO iteration, the amount of time steps allocated to training NFSP meta-NE solver in the new iteration is the previous amount multiplied by a coefficient of 1.5.

When solving the normal-form restricted game in PSRO, we calculate the meta-NE by running FP for 2000 iterations on the empirical payoff metric between population policies. To calculate the empirical payoff matrix, for each policy combination, we play 3000 games for $m$-Clone Leduc or 1000 games for the Loss Game. Empirical payoff matrix evaluations for PSRO are not counted when measuring experience used to train.

## 6.3 Training Hyperparameters

For 20-Clone Leduc, we used the OpenSpiel Leduc NFSP implementation as a starting point for our hyperparameters. We then adjusted our learning rate, batch size, and rollout steps to allow better wall-time speed through parallelization while maintaining roughly the same sample efficiency as the OpenSpiel reference. The same hyperparameters are used for the 20-Clone Leduc NXDO meta-NE solver and NFSP on the base 20-Clone Leduc game.

When training for the Loss Game, aside from the network architecture and epsilon greedy annealing period, the same NFSP hyperparameters were reused when training the NXDO meta-NE solver. Because the binned discrete action Loss Game is much more complex than the induced NXDO restricted game, new DDQN hyperparameters were found for NFSP on the original game using a random search of 50 samples over a space defined in table 8. DDQN hyperparameter trials were performed by training and measuring performance against a fixed adversary.

DDQN uses epsilon-greedy exploration. For 20-Clone Leduc experiments, in NFSP (both on the base game and restricted game), the epsilon exploration parameter is linearly annealed from 0.06 to 0.001 over 20e6 timesteps. In PSRO and NXDO, where BRs train over shorter periods, epsilon is linearly annealed from 0.20 to 0.001 over 1e5 timesteps. For NFSP in the Loss Game (both on the base game and restricted game) the epsilon exploration parameter is annealed from 0.06 to 0.001 over 500e6 timesteps. For all DDQN uses, learning does not start until 16,000 steps have been collected in the circular replay buffer.

Hyperparameters for Loss Game PPO BRs used by NXDO and PSRO were found in a random search of 50 samples over a space defined in table 9. Like DDQN, PPO hyperparameter trials were performed

by training and measuring performance against a fixed adversary. The same hyperparameters were used in both variations of the Loss Game.

Approximate exploitability in the Loss Game was measured by training a continuous-action PPO BR against each meta-NE (for NXDO and PSRO) or average policy (for NFSP) checkpoint and measuring the final average reward. These BRs share the same hyperparameters and stopping conditions as those used to train NXDO and PSRO.

Tables 1 through 7 display parameters used. Our deep RL implementations for NFSP, PSRO, and NXDO are built on top of the open source RLlib framework. [4]. Any parameters not listed were set to the RLlib 1.0.1 default values.

All RL algorithms use the Adam optimizer [2] with $\epsilon$ set to 1e-8 and 4 parrallel experience workers with 1 environment per worker. A discount factor of 1.0 is used in all games. For 20-Clone Poker, all algorithms used a fully-connected network with two layers of size 128 and relu activations. For the Loss Game, two layers of size 32 and tanh activations were used.

| circular buffer size | 2e5 |
|---|---|
| total rollout experience gathered each iter | 1024 steps |
| learning rate | 0.01 |
| batch size | 4096 |
| TD-error loss type | MSE |
| target network update frequency | every 10,000 steps |

Table 1: 20-clone Leduc DDQN hyperparameters (Used by NXDO BRs and meta-NE solver, PSRO BRs, base game NFSP)

| RL learner params | DDQN, see Table 1 |
|---|---|
| anticipatory param | 0.1 |
| avg policy reservoir buffer size | 2e6 |
| avg policy learning starts after | 16,000 steps |
| avg policy learning rate | 0.1 |
| avg policy batch size | 4096 |
| avg policy optimizer | SGD |

Table 2: 20-clone Leduc NFSP hyperparameters (Used by base game NFSP, NXDO meta-NE solver)

| GAE $\lambda$ | 1.0 |
|---|---|
| entropy coeff | 0.01 |
| clip param | 0.2 |
| KL target | 0.01 |
| learning rate | 5e-4 |
| train batch size | 2048 |
| sgd minibatch size | 256 |
| num sgd iters on train batch | 30 |
| separate policy and value networks | Yes |
| continuous action range | [-1.0, 1.0] for each dim |

Table 3: Loss Game Continuous Action PPO hyperparameters (Used by NXDO BRs, PSRO BRs)

| circular buffer size | 2e5 |
|---|---|
| total rollout experience gathered each iter | 1024 steps |
| learning rate | 0.01 |
| batch size | 4096 |
| TD-error loss type | MSE |
| target network update frequency | every 10,000 steps |

Table 4: Loss Game NXDO meta-NE DDQN hyperparameters (Used by NXDO meta-NE solver)

| | |
|---|---|
| RL learner params | DDQN, see Table 4 |
| anticipatory param | 0.1 |
| avg policy reservoir buffer size | 2e6 |
| avg policy learning starts after | 16,000 steps |
| avg policy learning rate | 0.1 |
| avg policy batch size | 4096 |
| avg policy optimizer | SGD |

Table 5: Loss Game NXDO meta-NE NFSP hyperparameters (Used by NXDO meta-NE solver)

| | |
|---|---|
| circular buffer size | 1e5 |
| learning starts after | 16,000 steps |
| total rollout experience gathered each iter | 64 steps |
| learning rate | 0.007 |
| batch size | 4096 |
| TD-error loss type | MSE |
| target network update frequency | every 1e5 steps |

Table 6: Loss Game NFSP DDQN hyperparameters (Used by base game NFSP)

| | |
|---|---|
| RL learner params | DDQN, see Table 6 |
| anticipatory param | 0.1 |
| avg policy reservoir buffer size | 2e6 |
| avg policy learning starts after | 16,000 steps |
| avg policy learning rate | 0.07 |
| avg policy batch size | 4096 |
| avg policy optimizer | SGD |

Table 7: Loss Game NFSP hyperparameters (Used by base game NFSP)

| | |
|---|---|
| circular buffer size | {5e4, 1e5, 2e5} |
| total rollout steps gathered each iter | {16, 32, 64, 128} |
| learning rate | log-uniform([0.0001, 0.1]) |
| batch size | {1024, 2048, 4096} |
| target network update frequency | {1e3, 1e4, 1e5} |

Table 8: Loss Game DDQN hyperparameter search space (Used by base game NFSP)

| | |
|---|---|
| GAE $\lambda$ | {0.9, 1.0} |
| entropy coeff | {0.0, 0.001, 0.01, 0.1} |
| clip param | {0.1, 0.2, 0.3} |
| KL target | {0.001, 0.01, 0.1} |
| learning rate | {5e-2, 5e-3, 5e-4, 5e-5, 5e-6} |
| train batch size | {2048, 4096} |
| sgd minibatch size | {128, 256, 512, 1024} |
| num sgd iters on train batch | {1, 5, 10, 30, 60} |

Table 9: Loss Game Continuous Action PPO hyperparameter search space (Used by NXDO BRs, PSRO BRs)

# 7 Costs and Benefits of Extensive-Form Restricted Games

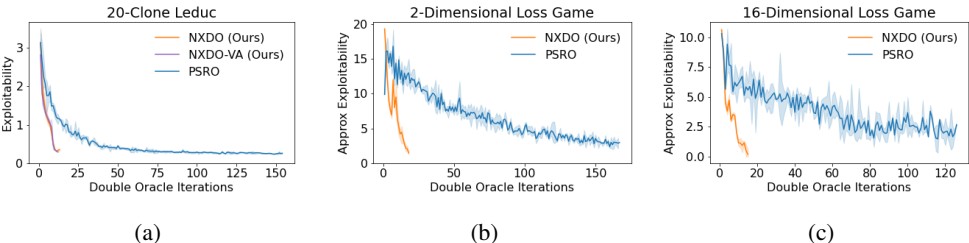

Figure 3: NXDO exploitability in 20-Clone Leduc (a) and approximate exploitability in the Loss Game (b and c) as a function of Double Oracle iterations in NXDO and PSRO.

Shown in figure 3, we compare the exploitability of NXDO against that of PSRO in terms of Double Oracle iterations. NXDO acheives a low exploitability in significantly less iterations due to mixing population strategies at every infostate in the game rather than just the at root like is done in PSRO. When both are limited to a small population of behavioral strategies, the extensive-form restricted game allows for much more expressive power in solving for a meta-NE than a normal-form restricted game does.

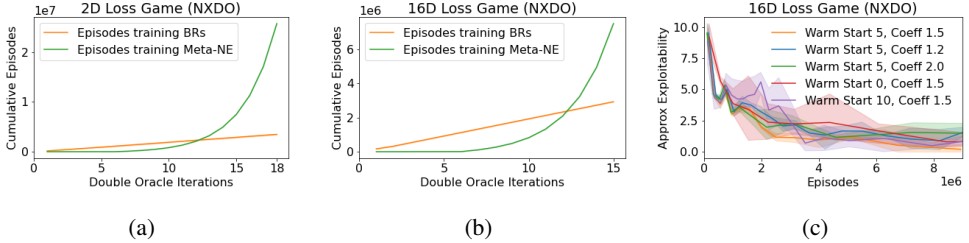

Figure 4: (a and b) In NXDO tests on the Loss Game, the cumulative amount of experience in episodes to either train BRs or calculate meta-NE with NFSP as a function of Double Oracle iterations. After the warm start phase, each NXDO iteration, we multiply the amount of episodes we spend training NFSP by a coefficient of 1.5. (c) NXDO ablations on the 16-Dimensional Loss Game. We vary the number of warm start iterations in which we spend zero time training the NFSP meta-NE solver and the coefficient by which we multiply the episodes spent training NFSP each iteration after the warm start.

This additional expressive power in the restricted game does come at an increased computational cost which needs to be taken into account. NXDO is most applicable when the need to mix population strategies at many infostates outweighs the extra computation needed to compute the extensive-form restricted game. Figures 4a and 4b show the cumulative episodes spent training either BRs or meta-NE vs NXDO iterations. The time spent training a meta-NE is gradually increased each iteration to trade quickly adding BRs for solving the meta-NE at a high accuracy.

# 8 Neural Methods on Kuhn and Leduc Poker

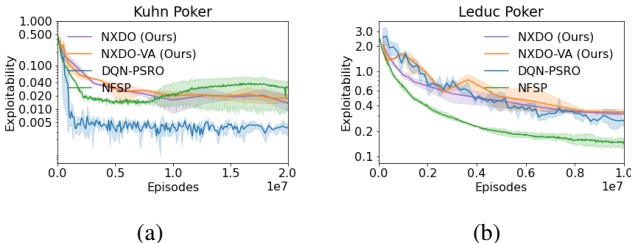

(a)                      (b)

Figure 5: NXDO, NXDO-VA, PSRO, and NFSP exploitability vs episodes collected on Kuhn and Leduc Poker

We test NXDO, NXDO-VA, PSRO, and NFSP on Kuhn and Leduc Poker using the same hyperparameters as used with 20-Clone Leduc. In these smaller games, NXDO performs less competitively because the time spent computing the extensive-form meta-NE strategies is large compared to the amount of time needed to train nearly all pure strategies for Kuhn and Leduc. NFSP also reaches an initial low exploitability quickly in part due to the small action-space sizes relative to other games tested.

# 9 Additional Loss Game Experiments

We analyze the effect of different stopping condition schedules for the NFSP meta-NE solver when training NXDO on the 16-Dimensional Loss Game. In figure 4c we compare a variaty of warm start amounts (the number of initial iterations in which zero time is spent training the meta-NE) and the coefficient at which we increase the number of episodes spent training any meta-NE thereafter. The default parameters are a warm start of 5 and a coefficient of 1.5, thus we spend 5 fixed NXDO iterations with a randomly initialized meta-NE solution, then train our sixth iteration meta-NE for the starting amount of 1e6 episodes, and then train each subsequent meta-NE for 1.5x the number of episodes in the previous iteration. Ablations with other reasonable values considered show that the fine details of such a schedule have only minor effects on performance in the Loss Game.

| | |
|---|---|
| NXDO against PSRO | $0.26 \pm 0.06$ |
| NXDO against NFSP | $1.28 \pm 0.06$ |
| PSRO against NFSP | $1.26 \pm 0.08$ |

Table 10: 2D Loss Game Round Robin Payoffs

| | |
|---|---|
| NXDO against PSRO | $4.43 \pm 0.06$ |

Table 11: 16D Loss Game Round Robin Payoffs

We also perform a round robin evaluation for the 2-Dimensional and 16-Dimensional Loss Game by measuring empirical payoffs between each algorithm's checkpoints after respectively 2e7 and 1e7 episodes of training. For a given matchup between two algorithms, we play 1000 games, swapping sides every other game, for each pairing of either algorithm's 3 seeds, resulting in 9000 games total. The reward means and 95% confidence intervals are reported in tables 10 and 11. NXDO beats PSRO and NFSP in our Loss Game evaluations.

## 10 Empirical Analysis of XDO Restricted Game Sizes

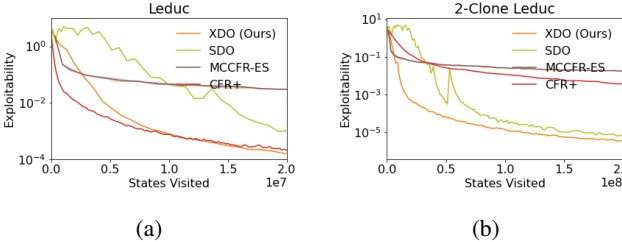

(a)            (b)

Figure 6: XDO exploitability compared to CFR$^+$, SDO and MCCFR as a function of states visited in (a) Leduc poker and (b) 2-Clone Leduc Poker.

XDO outperforms CFR$^+$ in games where Nash equilibria require mixing over a small fraction of actions. For example, XDO performs similarly to CFR$^+$ in standard Leduc poker, but XDO significantly outperforms CFR$^+$ in 2-Clone Leduc poker. XDO scales better in this transition because a restricted game sufficient to achieve a low exploitability can be induced with only a portion of the game's actions. Such a restricted game is much smaller in terms of total game states and thus requires less computation to solve. The number of states in 2-Clone Leduc poker is roughly 50 times that of standard Leduc poker, but when moving from standard to 2-Clone Leduc poker, the number of states in the XDO restricted game increases by a much smaller proportion.

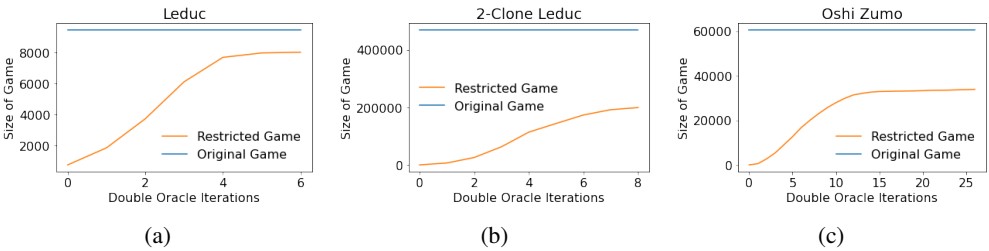

(a)          (b)          (c)

Figure 7: The size, in states, of the restricted game induced by the BR population in XDO in various games.

In our XDO experiments, the final size (in states) of the restricted game in Leduc Poker is roughly 85% of the size of the full game. The final size of the restricted game in 2-Clone Leduc Poker is only approximately 43% that of the full game, so XDO performs proportionally less computation to solve the restricted game meta-NE than it would take to directly solve the full-game NE with all actions considered. XDO also outperforms CFR in Oshi Zumo, where the final size of the restricted game is only half that of the full game.

## 11 Computational Costs

Experiments were performed a shared local computer with 128 CPU-cores, 2 RTX 3090 GPUs, and 256GB of RAM. Due to small network sizes, most neural experiments were performed without GPU acceleration. Neural experiments on 20-Clone Poker and the Loss Game used 8 to 16 cores each and took between 2 and 4 days to complete using 10 to 40GB of RAM. Tabular XDO experiments were run for up to 1 day, using a single core each and between 1 to 10GB of RAM.

## 12 Experiment Code

Code for tabular experiments can be found at `https://github.com/indylab/tabular_xdo`

Code for neural experiments can be found at `https://github.com/indylab/nxdo`