# OpenReview forum: "XDO: A Double Oracle Algorithm for Extensive-Form Games"
_NeurIPS.cc/2021/Conference — NeurIPS 2021 Poster_

### Official Review · Reviewer_ae9r · 2021-06-30

**Rating:** 5
**Confidence:** 5

**Summary:**

This paper introduces XDO and NXDO, algorithms for solving sequential imperfect-information games that are intended to be more scalable for games with very large action spaces. XDO starts by including only a small number of actions in every infostate and alternates between solving the constrained game and adding full-game best response actions to the constrained game. In the limit XDO is guaranteed to converge, and requires a number of iterations that is linear in the number of infostates (rather than exponential, which standard double oracle requires).

NXDO is a related algorithm that uses neural network function approximation. The algorithm maintains a population of candidate policies. At each infostate, the algorithm choose a probability distribution over these candidate policies. Once an equilibrium is computed in this restricted pool of policies, a full-game best response is computed and added to the pool. NXDO also converges, though it could potentially be much slower than traditional solvers because the number of candidate policies could grow extremely large.

**Limitations And Societal Impact:**

The lack of discussion and experiments regarding limitations of NXDO (and to a lesser extent XDO) is the biggest weakness of the paper. The paper would benefit from a greater investigation of the class of games where NXDO hurts performance and where it helps.

**Main Review:**

## XDO

XDO appears to be quite similar to related work in [1, 2, 3]. The authors highlight the similarity to [1] but not the other work. In particular, SAEF in [2] is quite similar. SAEF starts with a small action abstraction of a game and then repeatedly alternate between solving the abstracted game using CFR and adding high-regret full-game actions to the abstraction.

Throughout the XDO section the authors compare XDO to PSRO, but given that this is a tabular setting with exact best responses, is there any difference between PSRO and DO?

Regarding Figure 3, a lot seems off in these plots. It would be helpful if these plots were in log-log to make the performance easier to see. For Figure 3a, why are you comparing to XFP rather than CFR? For 3b and 3c, what form of CFR are you using? Are you using alternating updates? The convergence rate seems unusually slow. Also, what parameters are you using for XDO?

## NXDO

The authors clearly show that NXDO is beneficial in certain games. However, the paper does a poor job of describing when NXDO is useful and when it is detrimental. The one discussion of this is when the authors say:
> a drawback of meta-actions that delegate actions to population policies is that, as in PSRO, the number of meta-actions grows linearly with the number of iterations. This can eventually make the restricted game harder to solve than the original game. In our experiments, however, NXDO achieves significant improvements in exploitability within a very small number of iterations, such that the issue of action delegation does not becomes an obstacle.

This seems to be a function of the games you are testing on though. What happens if you run NXDO on Leduc, for example? What about Leduc with 10 cardsin the deck? I'd expect NXDO to do much worse than NFSP in those cases. In general, I think for a technique like this where it can perform quite poorly in some settings, it's important to present results that clarify where it is useful and where it is harmful. The paper currently doesn't do that and only show experiments where NXDO helps. This is my biggest criticism of the paper.

Intuitively it feels like there should be a way to get the benefits of XDO in a neural algorithm without the shortcomings of NXDO in games like Leduc. The paper appears to briefly discuss this with the mention of NXDO-VA. However, the description of NXDO-VA is very terse and I do not think I understand what the algorithm is.

## Overall

Given the existence of work quite similar to XDO, I feel that there is not much novelty to XDO and a lot of the novelty in the paper is actually in NXDO. However, NXDO has a major weakness that it could perform much worse than alternative algorithms in games with small action spaces (such as presumably Leduc poker, which only ever has at most 3 actions). The paper would benefit from a thorough investigation where NXDO is helpful and where it is harmful. Ideally, the authors would be able to come up with an algorithm like NXDO that is helpful, or at least not harmful, in all settings.

[1] Branislav Bošansky et al. An exact double-oracle algorithm for zero-sum extensive-form games with imperfect information. Journal of Artificial Intelligence 2014.

[2] Noam Brown and Tuomas Sandholm. Simultaneous Abstraction and Equilibrium Finding in Imperfect-Information Games. IJCAI 2015.

[3] Jiří Čermák, Branislav Bošansky, Viliam Lisy. An algorithm for constructing and solving imperfect recall abstractions of large extensive-form games. IJCAI 2017.

**Time Spent Reviewing:**

~5 hours

---

> ### Author Response · Authors · 2021-08-10
> **Author Response to ae9r**
>
> Thank you for the detailed and helpful review. You make a good point that it would be beneficial to include more results for NXDO. Here we have included results on Leduc, Leduc with 10 cards, Kuhn, and the 2-dimensional loss game with different discretization for NFSP (and continuous actions for NXDO and PSRO). Perhaps surprisingly, in each of these experiments, both versions of NXDO perform competitively with NFSP, although they do perform slightly worse in the poker games. The results are at this link:
>
> https://i.ibb.co/qMRRHm5/ae9r-nxdo-experiments.png
>
> A well-known downside to double oracle algorithms is that in many games they could take longer to complete than simply solving the game outright. This is because it could take many iterations for the double oracle algorithm to add best responses and solve the restricted game. If the ending restricted game is close in size to the original game, it would have been better to simply solve the original game. Despite this downside, there are two reasons why double oracle algorithms are popular approaches. First, in many games, the ending restricted game is much smaller than the original game, so a double oracle approach will end up being much faster than outright solving the original game. Second, many large games are too large to be solved outright, so some sort of abstraction must be made. Double oracle algorithms can be seen as a way of building up an abstracted game by adding actions to the restricted game. Our method, being a double oracle method, inherits these same drawbacks and advantages.
>
> We hypothesize that NFSP should perform well with respect to NXDO in games where there is a manageable number of actions and a large percent of them need to be mixed over to find a Nash (such as the 2D loss game with 100 actions, and Kuhn and Leduc poker). We could even construct games where NXDO (or any double oracle method) performs arbitrarily worse than NFSP (or another method such as Deep CFR) by ensuring that all Nash equilibria have full support at every infostate. As the size of these games increases, NXDO will do increasingly worse than NFSP. Conversely, NFSP will perform poorly relative to NXDO in domains that have a high number of actions (such as the 2-D loss game with 1,000 actions) but where an approximate Nash can be found with a smaller subset of actions.
>
> Importantly, NXDO is the only existing method that can handle high-dimensional continuous action games (for example, the 16-dimensional loss game). These games cannot be discretized due to the curse of dimensionality, so methods such as NFSP and Deep CFR cannot be used. The only other existing methods that can be used on these games are self play and PSRO. Self play is well known to not converge to a Nash and often cycles. PSRO, as shown in our results, does not converge to an approximate Nash equilibrium in the 16-dimensional loss game. Approximately solving large high-dimensional continuous action games is an important open problem. These games are quite common in applications such as robotics and robust reinforcement learning. Although on games like Leduc we perform slightly worse than NFSP, our method is the current state of the art method for large high dimensional continuous action games, which are domains where NFSP cannot be used. We did not adequately convey this in the paper and will edit the final version to better communicate this important fact of NXDO.
>
> To summarize, we believe that these additional experiments and discussion, which will be added to the final version of the paper, directly address your main concern. Please let us know if there are other experiments that are needed to better address your main concern and we may be able to add them.
>
> ### Additional comments
> Being a double oracle algorithm, our work is most similar to SDO. But because SDO considers populations of sequences of actions, even if one were to find best responses via deep RL, the size of the restricted game is restricted to the set of compatible sequences of actions. In large games, this restricted game will clearly grow too slowly to be practical. Please see our response to reviewer 6vBM for an in-depth discussion of the difference between the two methods.
>
> The other two methods you mentioned, while similar at a high level, are quite different in practice because neither method is a double oracle method. While Brown and Sandholm’s method does add actions every iteration, the way that actions are added is not through a best response. FPIRA, while it does add best responses, considers imperfect recall abstractions, which is a different approach from ours. Thank you for pointing out these two related works that we missed; we will include these references in the final version with a discussion about the differences between these works and our work.
>
> We will clarify what NXDO-VA does in detail in the final version.  When constructing the NXDO-VA restricted game, we do so in the same way as in tabular XDO. Like with XDO, the NXDO-VA restricted game action space is the same as the base game action space, where the only legal actions at a given infoset are those that would be chosen by a member of the population at that infoset. In contrast, the NXDO restricted game construction is different from tabular XDO, as the NXDO restricted game action space constitutes a selection of population policies with which to delegate the actual action decision to. The NXDO action space size is equal to the number of strategies in the population, whereas the NXDO-VA/XDO action space is that of the base game, where the legal actions at each infoset are the union of what each (deterministic) member of the population would choose. In NXDO, each action selects a member of the population to act on behalf of the restricted game agent, whereas in NXDO-VA and XDO, each action is directly applied to the base game.
>
> NXDO-VA exists to address a potential pitfall of NXDO, when the size of the NXDO restricted game action space size could grow to exceed the size of the base game action space resulting in a new restricted game that is unnecessarily difficult to solve. The NXDO-VA restricted game action space doesn’t change size over iterations like with NXDO, so it can’t suffer from the same issue. For this reason, NXDO is best suited for games with large or continuous action spaces.
>
> There is no difference between tabular PSRO and DO. Thank you for pointing this out. In the final version we will change PSRO to DO.
>
> We use OpenSpiel’s vanilla CFR, with alternating updates.
>
> We do not compare to CFR in figure 3.a because this chart only compares algorithms that use best responses and compares in terms of iterations on the x axis. A comparison to CFR on Leduc can be found in figure 6.a in the supplementary materials. We compare to CFR in terms of infostates expanded. In figure 3.b we also compare to CFR on the 2-clone Leduc game. The performance is slower than you might expect because 2-clone Leduc is a large game compared to Leduc. We will change all the plots to be in log-log to make them easier to see. Here we include additional experiments on Leduc, 2-clone Leduc and Oshi-Zumo with CFR+. As you predicted, CFR+ performs better than CFR, but when included in XDO the relative performance between XDO-CFR+ and CFR+ is the same as the difference between XDO-CFR and CFR. In particular, XDO and CFR perform about the same on Leduc and Oshi-Zumo, while XDO performs better than CFR on 2-clone Leduc.
> https://i.ibb.co/J72g29Y/ae9r-cfr-experiments.png

---

> > ### Comment · Area_Chair_Umwu · 2021-08-13
> > **Sufficient comparison to SAEF?**
> >
> > (this comment is not visible to the authors)
> >
> > Given the relatively simple algorithm proposed in the paper, it seems like experiments should be very convincing in order for it to be strong enough for NeurIPS. To that end, it seems to me that they should have compared to SAEF as well, since it might be expected to perform well in the same types of games as here?

---

> > ### Comment · Reviewer_ae9r · 2021-08-18
> > **Re: Author Response**
> >
> > Thank you for the additional experiments and detailed response.
> >
> > >Perhaps surprisingly, in each of these experiments, both versions of NXDO perform competitively with NFSP, although they do perform slightly worse in the poker games.
> >
> > Perhaps I'm missing something, but I would argue that the difference between NFSP and NXDO is not "slight" in Leduc and 10-card Leduc. NXDO appears to converge to an exploitability 2x higher than NFSP in those games, or takes much much longer to reach the same level of exploitability.
> >
> > >A well-known downside to double oracle algorithms is that in many games they could take longer to complete than simply solving the game outright. This is because it could take many iterations for the double oracle algorithm to add best responses and solve the restricted game. If the ending restricted game is close in size to the original game, it would have been better to simply solve the original game.
> >
> > It's true that DO can be slower than simply solving the entire game upfront. However, there are two major factors that mitigate this. First, the restricted game will never be bigger than the full game. From what I understand, this is *not* the case with NXDO. This seems like a big difference to me, and I would argue that XDO and NXDO are actually quite different for this reason. Second, when an action is added through DO, it is possible in some techniques to warm start the solving of the expanded restricted game using the solution from the previous restricted game.
> >
> > >While Brown and Sandholm’s method does add actions every iteration, the way that actions are added is not through a best response.
> >
> > I don't think this is true. Section 5 in their paper discusses adding actions and says "The opponents must have some defined strategies following this action. We can calculate a best response against those strategies."
> >
> > >When constructing the NXDO-VA restricted game, we do so in the same way as in tabular XDO. Like with XDO, the NXDO-VA restricted game action space is the same as the base game action space, where the only legal actions at a given infoset are those that would be chosen by a member of the population at that infoset. In contrast, the NXDO restricted game construction is different from tabular XDO, as the NXDO restricted game action space constitutes a selection of population policies with which to delegate the actual action decision to. The NXDO action space size is equal to the number of strategies in the population, whereas the NXDO-VA/XDO action space is that of the base game, where the legal actions at each infoset are the union of what each (deterministic) member of the population would choose. In NXDO, each action selects a member of the population to act on behalf of the restricted game agent, whereas in NXDO-VA and XDO, each action is directly applied to the base game.
> >
> > So it sounds like NXDO-VA is somewhat between XDO and NXDO? Would it be tractable to run on a large game like Texas hold'em?

---

> > > ### Author Response · Authors · 2021-08-20
> > > **Author Response**
> > >
> > > Thank you for the follow up comment.
> > >
> > > > Perhaps I'm missing something, but I would argue that the difference between NFSP and NXDO is not "slight" in Leduc and 10-card Leduc. NXDO appears to converge to an exploitability 2x higher than NFSP in those games, or takes much much longer to reach the same level of exploitability.
> > >
> > > Agreed, NFSP clearly outperforms NXDO in Leduc poker. Like all double oracle algorithms, NXDO will perform very well on some games and not well on others. We wish to focus our claims on games where NXDO performs well, such as games with large and continuous action spaces.
> > >
> > > > It's true that DO can be slower than simply solving the entire game upfront. However, there are two major factors that mitigate this. First, the restricted game will never be bigger than the full game. From what I understand, this is not the case with NXDO. This seems like a big difference to me, and I would argue that XDO and NXDO are actually quite different for this reason. Second, when an action is added through DO, it is possible in some techniques to warm start the solving of the expanded restricted game using the solution from the previous restricted game.
> > >
> > > Good point. While NXDO-VA will never make a restricted game larger than the full game, NXDO can. However, NXDO is intended for / works best in games with large action spaces, where this does not happen. As mentioned previously, NXDO is a very effective method for large high-dimensional continuous action games, and appears to be the state of the art on the loss surface game.
> > > As to your second point, would you please be able to provide a reference for this idea / suggestion? We are not aware of any DO methods that are able to warm start the solving of the expanded restricted game using the solution from the previous restricted game but this would indeed be an interesting direction to see if these methods can be combined with (N)XDO.
> > >
> > > > I don't think this is true. Section 5 in their paper discusses adding actions and says "The opponents must have some defined strategies following this action. We can calculate a best response against those strategies."
> > >
> > > It seems like using best responses in SAEF is proposed as a possible method to estimate regret growth rates of infostates outside of the abstraction. The best response actions are not necessarily added to the abstraction like in double oracle. It also appears that this method of using best responses was not actually used in the experiments. Additionally, this paper is a model-based approach, and the method does not seem like it can be easily extended to a neural version for large games with high-dimensional continuous actions.
> > >
> > > > So it sounds like NXDO-VA is somewhat between XDO and NXDO? Would it be tractable to run on a large game like Texas hold'em?
> > >
> > > Yes, NXDO-VA is somewhat between XDO and NXDO. NXDO-VA is better suited than NXDO for domains with very small action spaces, like Kuhn poker, where NXDO runs the risk of adding more best responses than actions. However, NXDO-VA requires sampling from each neural network in your population at every infostate you encounter while NXDO will only sample from one neural network, so NXDO-VA will become intractable whenever you have a large number of neural networks in your population. On the other hand, NXDO will not become intractable with a large number of BRs in the population, but it will become harder to solve the restricted game. We do not know if a prohibitively large number of policies would be necessary for NXDO-VA in Texas hold ‘em, but we expect that NXDO would do well in Texas hold ‘em because the NXDO restricted game constructed from delegate policies would still likely have a much smaller action space than the base game with unabstracted betting.
> > >
> > > NXDO may outperform PSRO on Texas hold ‘em because in large games, PSRO would have to mix over a very large number of best responses at the root of the game to find an approximate Nash (see Section 4 of the paper). We think that we could outperform NFSP on Texas hold ‘em with a large number of betting actions because we find that NFSP tends to scale poorly as the number of actions increases. With that being said, with a certain bet size abstraction, we wouldn’t be surprised if NFSP matches or even outperforms NXDO. We will try to include Texas hold ‘em results in the final version, but it might take too long.
> > >
> > > Here is an experiment we ran earlier comparing NFSP (red), PSRO (green) and NXDO (blue) on Leduc poker with a stack size of 1,000 (1,000 different raise actions): https://i.ibb.co/vqMcy6f/1000-leduc-approx-exploitability.png. The y axis is approximate exploitability measured by training a PPO BR to saturation against each checkpoint. We found that NXDO outperformed NFSP in this experiment and was competitive with PSRO. If we increased the game size to Texas hold ‘em, we would expect NXDO to scale better than PSRO.

---

> > > > ### Comment · Reviewer_ae9r · 2021-08-20
> > > > **Re: Author Response**
> > > >
> > > > >As to your second point, would you please be able to provide a reference for this idea / suggestion? We are not aware of any DO methods that are able to warm start the solving of the expanded restricted game using the solution from the previous restricted game but this would indeed be an interesting direction to see if these methods can be combined with (N)XDO.
> > > >
> > > > This is done in the Brown and Sandholm paper mentioned above. Warm starting is also common when solving the restricted game as a linear program or mixed integer program.
> > > >
> > > > >It seems like using best responses in SAEF is proposed as a possible method to estimate regret growth rates of infostates outside of the abstraction. The best response actions are not necessarily added to the abstraction like in double oracle. It also appears that this method of using best responses was not actually used in the experiments. Additionally, this paper is a model-based approach, and the method does not seem like it can be easily extended to a neural version for large games with high-dimensional continuous actions.
> > > >
> > > > It's true that the paper used an alternative heuristic for choosing the actions to add, though it seems straightforward to use BR instead. I think it's also true that a neural version of the algorithm is non-trivial. I pointed to it as related to XDO, not NXDO.

---

> > > > > ### Author Response · Authors · 2021-08-24
> > > > > **Author Response**
> > > > >
> > > > > Thank you for pointing this out. Yes, using a BR instead in SAEF would be very similar to a double oracle algorithm with warm starting. We are not aware of warm starting being used for double oracle algorithms but this is a very interesting future direction. We will look into using warm starting for XDO.

---

> > > > > > ### Author Response · Authors · 2021-09-02
> > > > > > **Warm Starting for XDO Follow Up**
> > > > > >
> > > > > > Thanks again for pointing out the possibility of warm-starting the expanded restricted game: we think this is a very promising direction of research for XDO.  However, it seems to us that the techniques described in the SAEF paper do not apply to double oracle algorithms like XDO in general.  The “adding actions to abstractions” method described in the paper seems to apply specifically to adding actions which lead only to completely new IISG (i.e. all descendants of the new state are in new infosets for all players, none of which were in the existing restricted game).  It’s not clear to us how (or if) the technique can be modified to warm-start a new action which leads to a new state which is part of a player’s infoset that’s already part of the restricted game.
> > > > > >
> > > > > > Is this right, or did we miss something in the paper?  If you have any additional references that would be convenient for you to provide with regards to warm starting after adding an action (either regret minimization or LP/MIP), we would also greatly appreciate those.  Thank you.

---

> > > > > > > ### Comment · Reviewer_ae9r · 2021-09-02
> > > > > > > **Warm starting**
> > > > > > >
> > > > > > > [1] might answer your question. It's for warm starting CFR, but warm starting XFP should be strictly easier.
> > > > > > >
> > > > > > > [1] Strategy-Based Warm Starting for Regret Minimization in Games. Brown & Sandholm. AAAI 2016.

---

> > > > > > > > ### Author Response · Authors · 2021-09-02
> > > > > > > > **Warm Starting Response**
> > > > > > > >
> > > > > > > > Thanks, we did look at that paper, but we found that it only discusses warm-starting CFR from an existing policy that contains action probabilities for all (infostate, action) pairs in the game we wish to solve (either by starting with an existing policy for the game, or by starting with a policy for an abstracted version of the game and then mapping each infostate in the unabstracted game to an infostate in the abstracted game).
> > > > > > > >
> > > > > > > > However, when adding actions in a double oracle algorithm, we could have (infostate, action) pairs where the infostate is not present in the previous restricted game (and we see no obvious method to map these to infostates in the previous restricted game), and we will also have (infostate, action) pairs where the action isn’t present in the previous restricted game. It’s not clear from our reading of this paper if there’s any method to assign probabilities to these (infostate, action) pairs which results in warm-starting performing better than not warm-starting.
> > > > > > > >
> > > > > > > > We remain very interested in this direction of research, but from our searching, we haven’t been able to find any previous work describing warm-starting after adding an action (either regret minimization or LP/MIP), other than SAEF.  If you have any additional references for doing so, we would greatly appreciate those. Thank you again for your time.

---

> > > > > > > > > ### Comment · Reviewer_ae9r · 2021-09-02
> > > > > > > > > **Warm starting**
> > > > > > > > >
> > > > > > > > > When you add an action through double oracle the probability for the added action is zero, so you can set the policy below that action arbitrarily and it won't affect the warm starting. For example, you could just set the policy to uniform random, so long as the probability for the added action is zero. This only leads to potential problems when the policy for the *other* player is also unknown below the added action, but that's only the case when you're adding a whole new IISG in which case you'd want to use something like SAEF.

---

> > > > > > > > > > ### Author Response · Authors · 2021-09-03
> > > > > > > > > > **Warm Starting**
> > > > > > > > > >
> > > > > > > > > > Thank you for the detailed discussion on this point. We will try this out in future work but unfortunately we do not have time to run these experiments before the discussion deadline. We will be sure to include discussion about this topic in the final version of the paper. Thank you again for your previous feedback; we will incorporate all of the additional experiments and discussion in the final version of the paper.
> > > > > > > > > >
> > > > > > > > > > Regarding the point
> > > > > > > > > >
> > > > > > > > > > > “This only leads to potential problems when the policy for the *other* player is also unknown below the added action, but that’s only the case when you’re adding a whole new IISG in which case you’d want to use something like SAEF”:
> > > > > > > > > >
> > > > > > > > > > Aren’t there cases where we add actions that don’t add entire new IISGs but still contain some descendants whose opponent infosets are new?  For example, in Starcraft or fog-of-war chess or phantom tic-tac-toe, we add a new player 1 action `a` in infostate `I`.  In the descendant histories of `I`, as long as player 2 doesn’t observe that the action has been taken, then player 2 is still in infosets that were present in the original game, but once they take some action that reveals that player 1 must have taken action `a`, then they are in new infostates and thus their policy is undefined.  It seems that in this case, we would have potential problems using the method outlined in your previous comment, but we would also be unable to use something like SAEF.

---

> > > > > > > > > > > ### Comment · Reviewer_ae9r · 2021-09-03
> > > > > > > > > > > **Warm starting**
> > > > > > > > > > >
> > > > > > > > > > > >Aren’t there cases where we add actions that don’t add entire new IISGs but still contain some descendants whose opponent infosets are new?
> > > > > > > > > > >
> > > > > > > > > > > Maybe. I would need to see a real example.
> > > > > > > > > > >
> > > > > > > > > > > >It seems that in this case, we would have potential problems using the method outlined in your previous comment, but we would also be unable to use something like SAEF.
> > > > > > > > > > >
> > > > > > > > > > > You can still use the warm starting techniques mentioned above. When using [1], the regret for the player 2 infoset would just be zero. That's exactly what the regret would be if the added player 1 action was always in the game but was never chosen on any previous iteration. When warm starting player 1's regret, you would need to define the policy for the unreached player 2 infoset. That could be uniform random, or if you're using a policy network then you could just use the output of the policy net for that infoset. You could probably do even better with something similar to SAEF adapted to this kind of situation, but it wouldn't be necessary for warm starting.

---

> > > > > > > > > > > > ### Author Response · Authors · 2021-09-03
> > > > > > > > > > > > **Warm Starting**
> > > > > > > > > > > >
> > > > > > > > > > > > That makes sense, thanks. I'm still not sure that would work because it won't work for regret matching in normal form games with added actions, but I will try this out for XDO.

---

### Official Review · Reviewer_6vBM · 2021-07-12

**Rating:** 6
**Confidence:** 4

**Summary:**

This paper proposes a Double-Oracle adaptation for extensive-form games, called XDO.
Similar to how previous work, PSRO, can be viewed as an approximation to the normal-form Double-Oracle, they introduce Neural XDO (NXDO) as an approximation for XDO.
Authors prove and experimentally demonstrate on small games that this approach is more computationally efficient, as applying (an approximation of) Double-Oracle to extensive-form games may need many more iterations.


**Limitations And Societal Impact:**

A well-know drawback of using Double-Oracle algorithms is the cost of computing the (oracle) best responses. This limitation should be more discussed in the work and shown in the experiments (perhaps by stating the running time of the algorithms).

**Main Review:**

Authors propose XDO as a new algorithm, however I am not entirely convinced of its originality. The algorithm is very closely related to the sequence-form double-oracle (SDO) algorithm, which also claims improved efficiency for similar reasons as XDO does (i.e. small size of supports in each infostate in games). While they somewhat address this in related work, the difference is still unclear to me.

1) What is the difference between "sequence-form BR" and "extensive-form BR"? I am familiar only with best-response (irrespective of the representation form of the game). Given two restricted games, one for SDO and XDO, would these individual BRs yield different strategies in each case? Can you provide an example that would show the difference?
2) The SDO algorithm uses temporary leaves and assigns a value to them when the currently added sequences do not lead to a terminal history. Similar problem should probably arise for XDO, so I am not sure if the algorithm is well defined for a game like in https://cw.fel.cvut.cz/b201/_media/courses/be4m36mas/mas_l07_beyond_efgs_2020.pdf slides 8-13. Problem is outlined at slide 12. What value would XDO assign to the temporary leaf, so that the restricted game is well defined and can be solved?
3) How does XDO deal with chance nodes? Are all actions of chance added to the restricted game? If yes, then I do not think there is a difference between the SDO and XDO algorithms.
4) If the algorithms are indeed "fundamentally different" (L131), please explain why one should prefer XDO vs SDO to make an approximate version of the algorithm (i.e. NXDO vs "neural SDO").

The experimental results with NXDO on small games look very promising for solving large games. Is there a reason why authors did not run NXDO on more challenging games?

Overall, I like this research direction. If I misunderstood the difference between XDO and SDO and authors provide convincing answers to the questions and update the paper accordingly, I am willing to raise the overall score.


More details:

- Proposition 2: I believe this claim should be made about Double-Oracle, not PSRO. Similar to how Algorithm 2 does not have a termination criterion, PSRO also does not necessarily terminate.
- L121: DREAM [25] and ARMAC [9] are model-free regret-based deep learning approaches.  -- so? What are authors trying to say?
- L360:  is be a -> is a

Supplementary:
- L59: all 6 pure strategies -> all 9 ?

**Time Spent Reviewing:**

5

---

> ### Author Response · Authors · 2021-08-10
> **Author Response to 6vBM**
>
> Thank you for the detailed and helpful review and thank you for pointing out that the difference between the XDO and SDO was not clearly expressed in our paper. We will include a version of the following discussion in the final version of the paper. Please let us know if anything remains unclear.
>
> The main difference between SDO and XDO, which we will clarify in the final revision of this paper, is that SDO considers sequences of actions to be members of the population while XDO considers full behavioral strategies to be members of the population. This difference (which we referred to as a difference between sequence-form best responses and extensive form best responses) leads to different ways of defining the restricted game as well. In particular, in SDO, sometimes two sequences are not compatible, in which case the restricted game must be truncated and a temporary leaf must be created and a value must be assigned to this leaf. In contrast, since XDO considers full behavioral strategies, every restricted game will have all leaf nodes in the restricted game be terminal nodes in the original game. This difference turns out to be crucial when extending these methods to large games with neural networks. In particular, the restricted game for SDO will end up growing at a much slower rate than the restricted game in XDO. As a result, a neural version of SDO will take much longer to terminate, and the strategy at most infostates will be the default strategy if terminated early.
>
> To illustrate the difference between the two algorithms, we provide an example run (linked below) of XDO on the same game that is presented as an example in section 4.1.3 in the SDO paper [1].
>
> Like in that paper, we represent actions that are in the restricted game by bold arrows. This example demonstrates how SDO creates smaller restricted games than XDO because it only considers infostates that can be reached by compatible sequences. In tabular games, SDO results in a cautious approach that only considers a small subset of infostates in order to prevent adding suboptimal actions to the restricted game. However, as we describe in the following section, this will cause major problems when trying to scale SDO to large games.
>
>
> # XDO vs. SDO Example
> The figure corresponding to XDO is at this link: https://i.ibb.co/yXvrftj/6vbm-xdo-game-tree.png
>
> The figure corresponding to SDO is at this link: https://i.ibb.co/WHjqD13/SDO.png and is also included as figure 4.1.3 in [1].
>
> Extensive-form pure strategies specify an action at every infostate. In this example we will refer to extensive-form pure strategies by concatenating the actions the strategy takes in every infostate. For example, the pure strategy for the circle player in step 1 in our diagram corresponds to ADFH. Sequence-form pure strategies specify a sequence of actions that must be internally consistent. For example, {∅, A, AD} is a valid sequence-form pure strategy but {∅, B, AD} is not.
>
> In step 0, both SDO and XDO start with an empty game tree. Let’s assume that the default strategy for both algorithms is uniform random.
>
> In step 1, both SDO and XDO add the same best responses to the default strategy for both players. However, SDO adds actual sequences of actions, in particular {∅, A, AD} for the circle player and {∅, y} for the box player. Since the AD sequence of actions for the circle player is not compatible with the y action for the square player, the restricted game is the game with A as the available action for circle and y as the available action for square. But since neither AE nor AF are in the sequence population, the restricted game in SDO at this step terminates in a temporary leaf at that point.
>
> In contrast, XDO adds full extensive form strategies, in this case adding ADFH for the circle player and y for the square player. Now the restricted game for XDO is as in step 1 in the diagram. Since XDO adds full extensive form strategies, there is no need to create a temporary leaf node. **After only one iteration, SDO and XDO result in a different restricted game. This is because SDO adds sequences of actions (best responses in sequence form) to the population while XDO adds a pure strategy defined at every infostate (best responses in extensive form). The SDO restricted game is much smaller than the XDO restricted game because SDO only considers infostates that can be reached by compatible sequences in the population.** There are three more steps for SDO, which are described in their paper. XDO has only two more steps, which are described in the figure.
>
> # Why XDO vs SDO?
> **XDO can be easily extended to a neural version via deep reinforcement learning. SDO cannot be easily extended to large games because the restricted game grows too slowly.**
>
> Let’s say we create a neural SDO where every iteration a deep reinforcement learning best response policy is trained against the meta-NE. Once this deep RL best response policy is finished training, it will play a sequence of actions that this policy takes to exploit the meta-NE. Now to create the restricted game, we must add these sequences of actions to our set of sequences and create a restricted game where we only consider nodes that are reached by some compatible combination of sequences of actions that exist in the population.
>
> One way to implement this would be to attach to each best response policy in the population the set of all infostates that it sees when it plays against the meta-NE. In large games, the set of infostates that can be stored in memory will be vanishingly small relative to the total number of infostates. Also, even if one were able to store all infostates that are compatible with each policy, in large games, the restricted game will contain many temporary leaf nodes, as in figure 3.b in Bosansky et. al. [1]. Since SDO is cautious and only adds infostates to the restricted game if a best response policy could play to reach that infostate against the meta-NE it was trained against, this restricted game will be much too small to work for large games. Because the restricted game will be so small, when transferred back to the original game, most actions will be default actions.
>
> Conversely, one can view NXDO as the same as neural SDO but where each policy is allowed to take actions on every infostate, even ones that it has not encountered during training. This means that XDO will sometimes add suboptimal off-policy actions to the restricted game in tabular games, but this difference is crucial when scaling to large games. Although SDO works well in some tabular games, a neural version will not work well in large games because it adds actions too conservatively and an inner-loop solver will not be able to generalize the restricted game to the full game.
>
> There are additional differences between the tabular algorithms. In general, XDO will tend to terminate in fewer iterations than SDO. This is because, as demonstrated above, XDO will usually expand more actions every iteration because each extensive form best response provides an action at every infostate. To answer your question about chance nodes: yes, XDO expands all actions of chance nodes in the restricted game but as described above, this does not make XDO the same as SDO.
>
> # Additional Comments
> > The experimental results with NXDO on small games look very promising for solving large games. Is there a reason why authors did not run NXDO on more challenging games?
>
> The high dimensional loss surface games with continuous actions are actually quite challenging games. Since CFR-based methods and NFSP require discrete actions, there currently exist only three methods that can train on large high-dimensional continuous-action games: self play, PSRO, and NXDO. Self play is not guaranteed to converge to a Nash equilibrium and usually cycles in interesting games. As discussed in the paper, PSRO must sample pure strategies at the root, which can lead to a large number of iterations before converging. Indeed, PSRO does not perform well in these loss surface games and does not converge to an approximate Nash. As shown in our results, NXDO is the only existing method that can achieve an approximate Nash equilibrium on these games. So even though these games may seem small, these games are actually quite challenging and no other method can solve them. To our knowledge, NXDO is not only state of the art on these games, but it is the first and only method that can achieve an approximate Nash equilibrium in these games. Additionally, we are currently running experiments with NXDO vs. PSRO on Mujoco Robosumo, in which two Mujoco humanoids try to push each other out of a ring using high-dimensional continuous actions. This is an extremely challenging environment and we currently have promising preliminary results.
>
> > Proposition 2: I believe this claim should be made about Double-Oracle, not PSRO. Similar to how Algorithm 2 does not have a termination criterion, PSRO also does not necessarily terminate.
>
> -----> Yes, thank you. We will change this to be DO instead of PSRO.
>
> > L121: DREAM [25] and ARMAC [9] are model-free regret-based deep learning approaches. -- so? What are authors trying to say?
>
> -----> We will rewrite this sentence to better describe these methods.
>
> > L360: is be a -> is a
>
> -----> Thank you, we will change this.
>
> Supplementary:
> > L59: all 6 pure strategies -> all 9 ?
>
> -----> This is actually correct as is. There are 6 column strategies and 9 row strategies.
> > A well-known drawback of using Double-Oracle algorithms is the cost of computing the (oracle) best responses.
> ---->We will edit the paper to more clearly discuss the cost of computing the best responses. We will also add running time comparisons in the experiments.
>
> [1] Branislav Bošansky et al. An exact double-oracle algorithm for zero-sum extensive-form games with imperfect information. Journal of Artificial Intelligence 2014.

---

> > ### Comment · Reviewer_6vBM · 2021-08-31
> > **Re: Author Response**
> >
> > Thank you for the explanation of the differences between XDO and SDO. The provided reasoning is convincing, please make sure to update the paper accordingly.

---

### Official Review · Reviewer_H8Fr · 2021-07-14

**Rating:** 6
**Confidence:** 5

**Summary:**

The authors propose Extensive-Form Double Oracle (XDO), an extensive-form double oracle algorithm that is guaranteed to converge to an approximate Nash equilibrium in two-player zero-sum extensive-form games. The algorithm extends ides introduced in PSRO to the extensive-form setting in a simple and natural way. In particular, XDO mixes best responses at every information state, which guarantees convergence in a number of iterations which is linear in the number of information states. Moreover, the authors show how to apply XDO within an RL-like framework where best responses at each iteration are estimated through deep RL techniques. The experimental evaluation shows that this approach is promising both in the tabular and in the "neural" version.

**Ethical Concerns:**

No ethical concerns

**Limitations And Societal Impact:**

See Main review

**Main Review:**

I think the paper makes a useful contribution in formalizing how to generalize PSRO to the extensive-form setting. Hovewer, since XDO is a natural extension of PSRO I think the technical contributions of the paper are not particularly exciting. The main value of the work would be giving a clear reference point for other reserachers who want to "safely" apply PSRO-like methods to extensive form games.

I think the experimental evaluation could be more convincing if the following items were addressed:

- The game instances used for plot  (a) and (b) are too small to be convincing, and they do not really help in evaluating which is the limit of scalability of XDO. Moreover, it would be better to include a baseling using RM+ (like CFR+).
- I'm not familiar with Oshi-Zumo so it's difficult to evaluate plot (c). Adding a remark on the dimensions of these games in terms of information states or sequences in the main paper would help when reading the experimental results. If one were ok with exploitability 10^-2, XDO would double the number of states visits.

Minor comments: I would add a remark on the players having perfect recall in Theorem 1.

**Time Spent Reviewing:**

4

---

> ### Author Response · Authors · 2021-08-10
> **Author Response to H8Fr**
>
> Thank you for your insightful comments. We agree that a main contribution of this paper is providing a sound way of extending PSRO to extensive form games. We would like to point out that another main contribution of this paper is providing a method, NXDO, which can achieve approximate Nash equilibria on large high-dimensional continuous-action games. Since CFR-based methods and NFSP require discrete actions, there currently exist only three methods that can train on large high-dimensional continuous-action games: self play, PSRO, and NXDO. Self play is not guaranteed to converge to a Nash equilibrium and usually cycles in interesting games. As discussed in the paper, PSRO must sample pure strategies at the root, which can lead to a large number of iterations before converging. Indeed, PSRO does not perform well in these loss surface games and does not converge to an approximate Nash. As shown in our results, NXDO is the only existing method that can achieve an approximate Nash equilibrium on these games. To our knowledge, NXDO is not only state of the art on these games, but it is the first and only method that can achieve an approximate Nash equilibrium in a reasonable amount of time on these games.
>
> As requested, we have new results on all games with CFR+, shown in the figures linked here (https://i.ibb.co/MGrwTXB/H8-Fr-cfr-experiments.png ), and will include them in the final version of the paper. 2-clone Leduc poker is actually a fairly large game, and Oshi-Zumo is even larger. We did not include larger games than Oshi-Zumo in our tabular experiments because we ran into memory constraints. For larger games such as the high-dimensional continuous-action loss game we did include neural experiments with NXDO. Here are the sizes of all tabular games, which we will include in a table in the final version:
>
> Kuhn: 58 states, 12 infostates
>
> Leduc: 9457 states, 936 infostates
>
> 2-clone Leduc: 468,517 states, 3,894 infostates
>
> Oshi Zumo (4 coins, 3 spaces on board, time horizon of 6): 60,553 states, 20,868 infostates

---

### Official Review · Reviewer_1Gab · 2021-07-19

**Rating:** 7
**Confidence:** 4

**Summary:**

This paper extends the PSRO family to efficiently run in extensive form games. This allows the convergence to be linear in number of infostates, in contrast to the potentially exponentially larger matrix game.


**Limitations And Societal Impact:**

Please address.

**Main Review:**

I think the paper is a nice and important addition to the PSRO algorithm family. I also appreciate that authors include non-tabular methods. The idea is relatively straightforward, as the algorithm simply expands the extensive form game tree and runs CFR (and there is existing work on DO methods that expand the EFG). The paper also mostly does a good job explaining the ideas.

The experiments are also non trivial and well done.

I have seen this paper before, and I think there are substantial improvements - more experiments and multiple fixes.

I also like the analysis through the k-GMP and m-clone GMP game

Comments:
> [Abstract] Experiments on a modified Leduc poker game show that tabular XDO achieves over 11x lower exploitability than CFR and over 82x lower exploitability than PSRO and XFP in the same amount of time.
  I am not sure this is supported by the experiments. The only related graph I see compares 2-Clone Leduc in terms of states visited, not time. I think states visited makes more sense, the appendix should just match the actual experiment.

Minor:
> [336] In figure -> In Figure
> [344] figures -> Figures


**Time Spent Reviewing:**

2

---

> ### Author Response · Authors · 2021-08-10
> **Author Response to 1Gab**
>
> Thank you for the helpful review. You are correct, in the experiments we compare on infostates expanded, not time. In an earlier version of the paper we compared based on time but did not update the abstract. We will update the abstract in the final version of the paper.
>
> We will also make the suggested capitalization edits in the final version of the paper.

---

### Decision · Program_Chairs · 2021-09-28

**Decision:**

Accept (Poster)

**Comment:**

This submission had a range of scores across reviewers, with votes both for reject and accept. However, there is actually a strong consensus on the strength and weaknesses of the paper. Conceptually, it's a fairly straightforward idea, but one that may be worth exploring in light of the performance of other double-oracle methods. However, all reviewers agreed that it is not a particularly novel direction.

The numerical strength of the approach is not very positive: the authors show good results on games that the authors designed in order to get good performance from their proposed algorithms. On the existing games that were tried, the proposed algorithms had fairly weak performance. At the same time, the reviewers all felt that the authors did a good job experimentally: they tried a good number of small and medium-sized games and show that XDO/NXDO performs poorly in certain settings and quite well in other (somewhat contrived) settings. Thus one could argue that the paper is a pretty good reference point for the method: perhaps the approach is not that useful overall, and this paper lays out that performance in a reasonable way.
The authors are strongly encouraged to include the additional experiments from the rebuttals.

**Consistency Experiment:**

NeurIPS has a long history of experimentation. In 2014, NeurIPS ran an experiment in which 10% of submissions were reviewed by two independent committees to quantify the randomness in the review process. This year, we repeated a variant of this experiment to see how the quality of the review process has changed over time.  This paper was part of the experiment and was therefore assigned to two committees (consisting of reviewers, an Area Chair, and a Senior Area Chair) that reached independent decisions.  If both committees made the same recommendation, this recommendation was followed. If a single committee recommended acceptance, the paper was accepted (with the exception of a few cases in which the other committee identified what we considered a fatal flaw, e.g., an error in a key result).

Both committees reached the same decision: **Accept (Poster)**

The other committee assigned to the paper recommended **Accept (Poster)**.  You can find the other set of reviews, along with any follow up discussion with the authors here:
https://openreview.net/forum?id=sMYnHv5I14V